# The cortical representation of language timescales is shared between reading and listening
Catherine Chen [1] ✉, Tom Dupré la Tour [2], Jack L. Gallant [2], Daniel Klein[1] & Fatma Deniz [2,3,4] ✉

Language comprehension involves integrating low-level sensory inputs into a hierarchy of increasingly high-level features. Prior work studied brain representations of different levels of the language hierarchy, but has not determined whether these brain representations are shared between written and spoken language. To address this issue, we analyze fMRI BOLD data that were recorded while participants read and listened to the same narratives in each modality. Levels of the language hierarchy are operationalized as timescales, where each timescale refers to a set of spectral components of a language stimulus. Voxelwise encoding models are used to determine where different timescales are represented across the cerebral cortex, for each modality separately. These models reveal that between the two modalities timescale representations are organized similarly across the cortical surface. Our results suggest that, after low-level sensory processing, language integration proceeds similarly regardless of stimulus modality.

Humans leverage the structure of natural language to convey complex ideas that unfold over multiple timescales. The structure of natural language contains a hierarchy of components, which range from low-level components such as letterforms or articulatory features, to higher-level components such as sentence-level syntax, paragraph-level semantics, and narrative arc. During human language comprehension, brain representations of low-level components are thought to be incrementally integrated into representations of higher-level components[1]. These representations have been shown to form a topographic organization across the surface of the cerebral cortex during spoken language comprehension[2–6].

Both written and spoken language consist of a hierarchy of components, but to date it has been unclear to what extent brain representations of these hierarchies are shared between the two modalities of language comprehension. At low levels of the hierarchy, brain representations are known to differ between the two stimulus modalities. For example, visual letterforms in written language are represented in the early visual cortex, whereas articulatory features in spoken language are represented in the early auditory cortex[7,8]. In contrast, many parts of temporal, parietal, and prefrontal cortices process both written and spoken language[9–14]. It could be the case that in these areas representations of higher-level language components are organized in the same way for both written and spoken language comprehension. On the other hand, these areas could contain overlapping but

independent representations for the two modalities. One way to differentiate between these two possibilities would be to directly compare the cortical organization of brain representations across high-level language components between reading and listening. However, prior work has not performed this comparison. Most prior studies of reading and listening have compared brain responses generally, without explicitly describing what stimulus features are represented in each brain area[9–12]. Other studies focused on relatively few components (e.g., low-level sensory features, word-level semantics, and phonemic features), and therefore did not provide a detailed differentiation between different levels of the language hierarchy[13,14]. Studies that did differentiate between different levels focused on one modality of language[2,5,6,15]. Prior studies are therefore insufficient to determine whether brain representations of the language hierarchy are organized similarly between reading and listening.

To address this problem, we compared where different levels of the language hierarchy are represented in the brain during reading and listening. Intuitively, levels of processing hierarchy can be considered in terms of number of words. For example, low-level sensory components such as visual letterforms in written language and articulatory features in spoken language vary within the course of single words; sentence-level syntax varies over the course of tens of words; paragraph-level semantics varies over the course of hundreds of words. Therefore we operationalize levels of the language

[1]Department of Electrical Engineering and Computer Sciences, University of California, Berkeley, CA, USA. [2]Helen Wills Neuroscience Institute, University of California, Berkeley, CA, USA. [3]Institute of Software Engineering and Theoretical Computer Science, Technische Universität Berlin, Berlin, Germany. [4]Bernstein Center for Computational Neuroscience, Berlin, Germany. ✉e-mail: cathychen@berkeley.edu; deniz@tu-berlin.de

hierarchy as language timescales, where a language timescale is defined as the set of spectral components of a language stimulus that vary over a certain number of words. For brevity, we refer to "language timescales" simply as timescales.

We analyzed functional magnetic resonance imaging (fMRI) recordings from participants who read and listened to the same set of narratives[13,16]. The stimulus words were then transformed into features that each reflect a certain timescale of stimulus information: first a language model (BERT) was used to extract contextual embeddings of the narrative stimuli, and then linear filters were used to separate the contextual embeddings into timescale-specific stimulus features. Voxelwise-encoding models were used to estimate the average timescale to which each voxel is selective, which we refer to as the "timescale selectivity". These estimates reveal where different language timescales are represented across the cerebral cortex for reading and listening separately. Finally, the cortical organization of timescale selectivity was compared between reading and listening.

## Results

We compared the organization of timescale representations between written and spoken language comprehension for each participant. First, the set of language-selective voxels for each modality was identified as those for which any of the timescale-specific language feature spaces significantly predicted blood oxygenation level-dependent (BOLD) responses (one-sided permutation test, $P < 0.05$, false discovery rate (FDR) corrected). Then, voxel timescale selectivity was compared between reading and listening across the set of voxels that are language-selective for both modalities. For each participant, voxel timescale selectivity is significantly positively correlated between the two modalities (S1: $r = 0.41$, S2: $r = 0.58$, S3: $r = 0.44$, S4: 0.34, S5: 0.47, S6: 0.35, S7: 0.40, S8: 0.49, S9: 0.52, $P < 0.001$ for each participant; Fig. 1a). Visual inspection of voxel timescale selectivity across the cortical surface confirms that the cortical organization of timescale selectivity is similar between reading and listening (Fig. 1b, c). For both modalities, timescale selectivity varies along spatial gradients from intermediate timescale selectivity in the superior temporal cortex to long-timescale selectivity in the inferior temporal cortex, and from intermediate timescale selectivity in the posterior prefrontal cortex to long timescale selectivity in the anterior prefrontal cortex. Medial parietal cortex voxels are selective for long timescales for both modalities. Estimates of timescale selectivity are robust to small differences in feature extraction—results are quantitatively similar when using a fixed rolling context instead of a sentence input context, and when using units from only a single layer of BERT instead of from all layers (Supplementary Figs. S1–S5). These results suggest that for each individual participant, representations of language timescales are organized similarly across the cerebral cortex between reading and listening.

In contrast to representations of language timescales, low-level sensory features are represented in modality-specific cortical areas. Figure 1d shows the prediction performance of linguistic features (i.e., timescale-specific feature spaces), and the prediction performance of low-level sensory features (i.e., spectrotemporal representations of auditory stimuli, and motion energy representations of visual stimuli). Voxels are colored according to the prediction performance of each set of feature spaces: voxels shown in blue are well-predicted by the linguistic feature spaces, voxels shown in orange are well-predicted by the low-level sensory feature spaces, and voxels shown in white are well-predicted by both sets of feature spaces. For both reading and listening, timescale-specific feature spaces predict well broadly across temporal, parietal, and prefrontal cortices. In contrast, low-level stimulus features predict well in the early visual cortex (EVC) during reading only, and in the auditory cortex (AC) during listening only. These results indicate that during language comprehension, linguistic processing occurs in similar cortical areas between modalities, whereas low-level sensory processing occurs in modality-specific cortical areas.

Within each participant, estimates of timescale selectivity depend not only on the presentation modality, but also on the presentation order. This is because each participant either read all the stories before listening to the

stories, or vice versa, and attentional shifts between novel and known stimuli may cause small differences in estimated timescale selectivity. Indeed, activation across higher-level brain regions is often more widespread and consistent for the first presentation modality than for the second presentation modality, indicating that participants attend more strongly to novel stimuli (Supplementary Figs. S6 and S7). In six of the nine participants, timescale selectivity was slightly longer for the first presented modality than for the second presented modality (Supplementary Fig. S8). This change in timescale selectivity between novel and repeated stimuli suggests that the predictability of high-level narrative components in known stimuli may reduce brain responses to longer language timescales. Nevertheless, the overall cortical organization of timescale selectivity was consistent between reading and listening across all nine participants, regardless of whether they first read or listened to the narratives. This consistency indicates that the effects of stimulus repetition on timescale selectivity are small relative to the similarities between timescale selectivity during reading and listening.

In order to consolidate results across all participants, we computed group-level estimates of timescale selectivity. To compute group-level estimates, first the estimates for each individual participant were projected to the standard FreeSurfer fsAverage vertex space[17]. Then, for each vertex, the group-level estimate of timescale selectivity was computed as the mean of the fsAverage-projected values. This mean was computed across the set of participants in whom the vertex was language-selective. Group-level estimates were computed separately for reading and listening. Group-level timescale selectivity was then compared between reading and listening across the set of vertices that were significantly predicted in at least one-third of the participants for both modalities (Supplementary Fig. S9 shows the number of participants for which each vertex was significantly predicted, separately for each modality). This comparison showed that timescale selectivity is highly correlated between reading and listening at the group level ($r = 0.48$; Fig. 2a). Cortical maps of group-level timescale selectivity (Fig. 2b) visually highlight that the spatial gradients of timescale selectivity across temporal and prefrontal cortices are highly similar between the two modalities. Gradients of timescale selectivity are also evident within previously proposed anatomical brain networks (Supplementary Fig. S10). Overall, these group-level results show that across participants, the organization of representations of language timescales is consistent between reading and listening.

The results shown in Figs. 1 and 2 indicate that timescale selectivity is similar between reading and listening. However, timescale selectivity alone is insufficient for determining whether representations of different timescales are shared between reading and listening—timescale selectivity could equate voxels with a very peaked selectivity for a single frequency band, and voxels with uniform selectivity for many frequency bands (Supplementary Fig. S11 shows how the uniformity of timescale selectivity varies across voxels). To investigate this possibility we used the timescale selectivity profile, which reflects selectivity for each timescale separately. Although the timescale selectivity profile is a less robust metric than timescale selectivity (see Methods for details), the timescale selectivity profile can distinguish between peaked and uniform selectivity profiles.

Figure 3 shows the Pearson correlation coefficient between the timescale selectivity profile in reading and in listening on the flattened cortical surface of each participant. The timescale selectivity profile is highly correlated between reading and listening across voxels that are language-selective in both modalities.

To further demonstrate the shared organization of cortical timescale selectivity, we compared the cortical distribution of selectivity for each of the eight timescales. For each of the eight timescales, we computed the correlation between selectivity for that timescale during reading and listening across the set of voxels that are language-selective in both modalities. The correlations for each timescale and participant are shown in Fig. 4a. A full table of correlations and statistical significance is shown in Supplementary Table S1. Selectivity for each timescale was positively correlated between reading and listening for each timescale and in each individual participant. Most of these correlations were statistically significant (one-sided

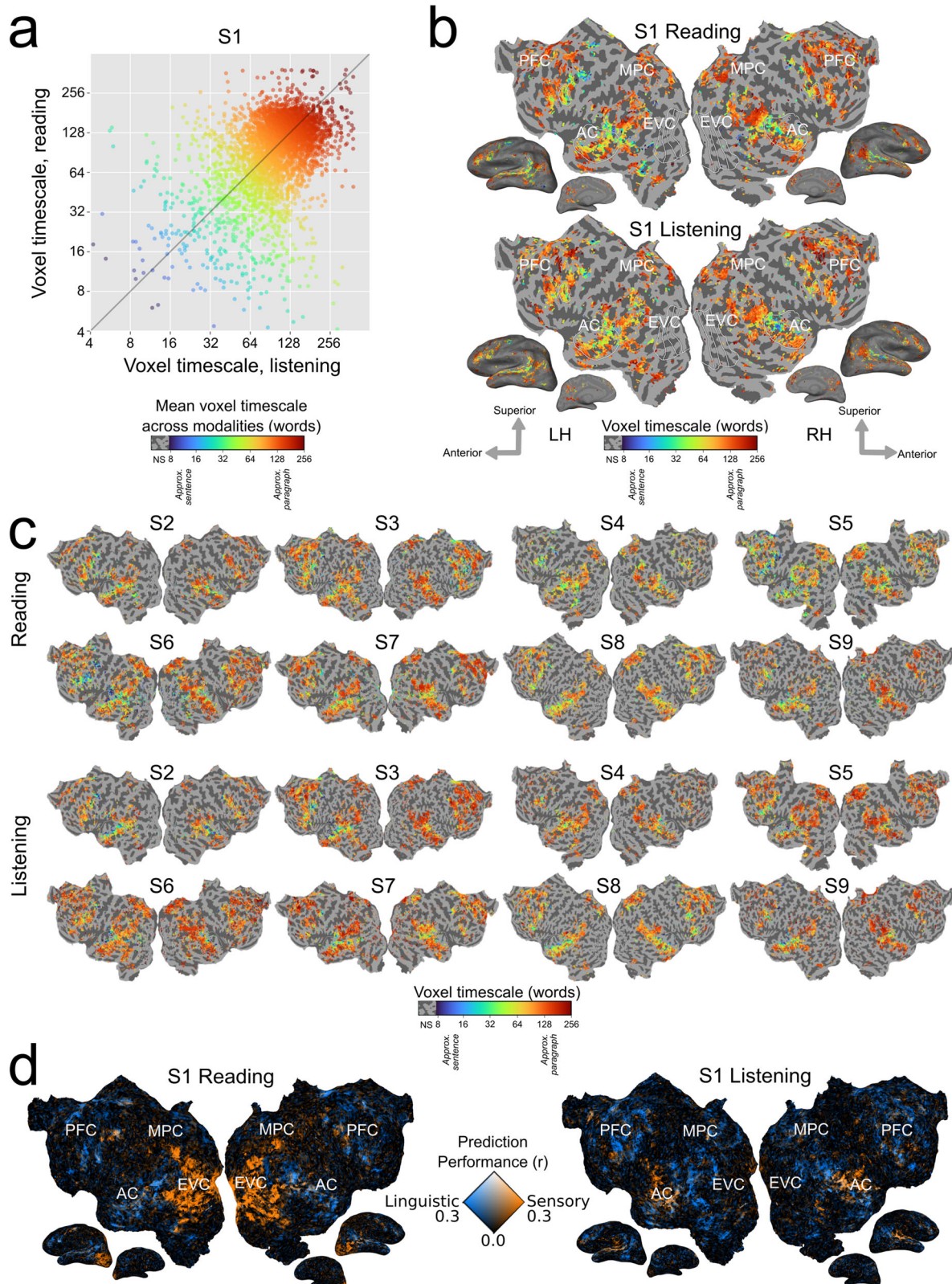

permutation test, $P < 0.05$, FDR-corrected). Note that comparing the timescale selectivity metric is more robust to noise in the data than comparing selectivity for each timescale separately (see Methods for details). Therefore, correlations between selectivity for each individual timescale are less consistent across participants than correlation between timescale selectivity.

The cortical distribution of selectivity for each timescale is shown for reading and listening separately in Fig. 4b. For concision, these results are shown at the group level. Visual inspection of Fig. 4b shows that for both reading and listening, short timescales (2–4 words, 4–8 words, 8–16 words) are represented in posterior prefrontal cortex and superior temporal cortex; intermediate timescales (16–32 words, 32–64 words) are represented

**Fig. 1 | Timescale selectivity across the cortical surface.** Voxelwise modeling was used to determine the timescale selectivity of each voxel, for reading and listening separately (see Methods for details). **a** Timescale selectivity during listening (*x* axis) vs reading (*y* axis) for one representative participant (S1). Each point represents one voxel that was significantly predicted in both modalities. Points are colored according to the mean of the timescale selectivity during reading and listening. Blue denotes selectivity for short timescales, green denotes selectivity for intermediate timescales, and red denotes selectivity for long timescales. Timescale selectivity is significantly positively correlated between the two modalities ($r = 0.41$, $P < 0.001$). Timescale selectivity was also significantly positively correlated in the other eight participants (S2: $r = 0.58$, S3: $r = 0.44$, S4: 0.34, S5: 0.47, S6: 0.35, S7: 0.40, S8: 0.49, S9: 0.52, $P < 0.001$ for each participant). **b** Timescale selectivity during reading and listening on the flattened cortical surface of S1. Timescale selectivity is shown according to the color scale at the bottom (same color scale as in (**a**)). Voxels that were not significantly predicted are shown in gray (one-sided permutation test, $P < 0.05$, FDR-corrected; LH left hemisphere, RH right hemisphere, NS not

significant, PFC prefrontal cortex, MPC medial parietal cortex, EVC early visual cortex, AC auditory cortex). For both modalities, temporal cortex contains a spatial gradient from intermediate to long-timescale selectivity along the superior to the inferior axis, the prefrontal cortex (PFC) contains a spatial gradient from intermediate to long-timescale selectivity along the posterior to the anterior axis, and precuneus is predominantly selective for long timescales. **c** Timescale selectivity in eight other participants. The format is the same as in (**b**). **d** Prediction performance for linguistic features (i.e., timescale-specific feature spaces) vs. low-level sensory features (i.e., spectrotemporal and motion energy feature spaces) for S1. Orange voxels were well-predicted by low-level sensory features. Blue voxels were well-predicted by linguistic features. White voxels were well-predicted by both sets of features. Low-level sensory features predict well in early visual cortex (EVC) during reading, and in early auditory cortex (AC) during listening. Linguistic features predict well in similar areas for reading and listening. After early sensory processing, cortical timescale representations are consistent between reading and listening across temporal, parietal, and prefrontal cortices.

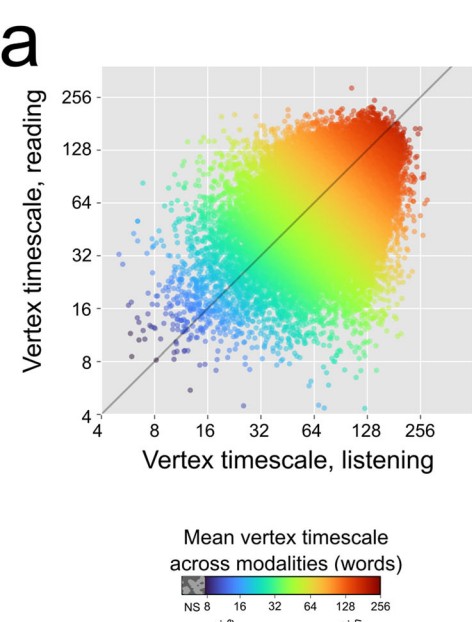

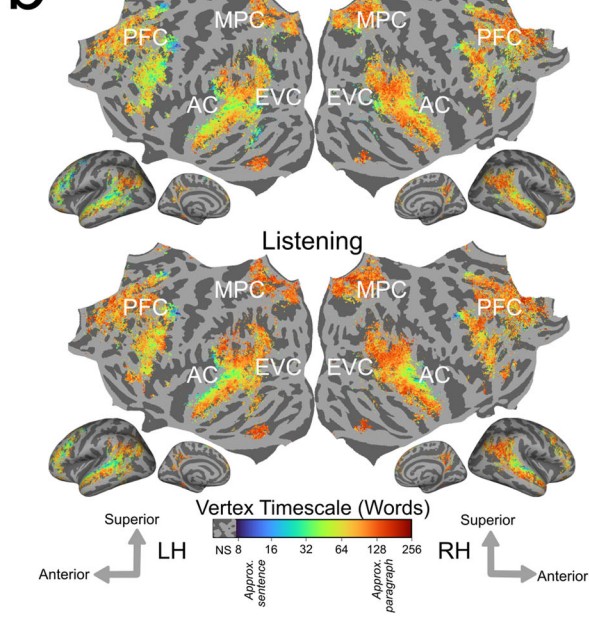

**Fig. 2 | Group-level estimates of timescale selectivity in standard brain space.** Group-level estimates of timescale selectivity are shown in a standard fsAverage vertex space. The group-level estimate for each vertex was computed by taking the mean over all participants in whom the vertex was language-selective. **a** Group-level timescale selectivity during listening (*x* axis) vs reading (*y* axis). Each point represents one vertex that was significantly predicted for both modalities for at least one-third of the participants. Each point is colored according to the mean of the group-level timescale selectivity during reading and listening. Blue denotes selectivity for short timescales, green denotes selectivity for intermediate timescales, and red denotes selectivity for long timescales. Timescale selectivity is positively correlated between the two modalities ($r = 0.48$). **b** For reading and listening separately group-level timescale selectivity is shown according to the color scale at the bottom (same

color scale as in (**a**)). Colored vertices were significantly predicted for both modalities in at least one-third of the participants. Vertices that were not significantly predicted are shown in gray (one-sided permutation test, $P < 0.05$, FDR-corrected; NS not significant, PFC prefrontal cortex, MPC medial parietal cortex, EVC early visual cortex, AC auditory cortex). Group-averaged measurements of timescale selectivity are consistent with measurements observed in individual participants (Fig. 1). For both modalities, there are spatial gradients from intermediate to long-timescale selectivity along the superior to the inferior axis of the temporal cortex, and along the posterior to anterior axis of prefrontal cortex (PFC). Precuneus is predominantly selective for long timescales for both modalities. Across participants, the cortical representation of different language timescales is consistent between reading and listening across temporal, parietal, and prefrontal cortices.

broadly across temporal, prefrontal, and medial parietal cortices; and long timescales (64–128 words, 128–256 words, 256+ words) are represented in prefrontal cortex, precuneus, temporal parietal junction, and inferior temporal cortex. The correlations between selectivity for each timescale and qualitative comparisons of the cortical distribution of selectivity for each timescale between reading and listening indicate that representations of language timescales are organized similarly between reading and listening.

## Discussion

This study tested whether representations of language timescales are organized similarly between reading and listening. We used voxelwise-encoding models to determine the selectivity of each voxel to different language timescales and then compared the organization of these representations between the two modalities (Fig. 5). These comparisons show that timescale selectivity is highly correlated between reading and listening across voxels that are language-selective in both modalities. This correlation is evident in

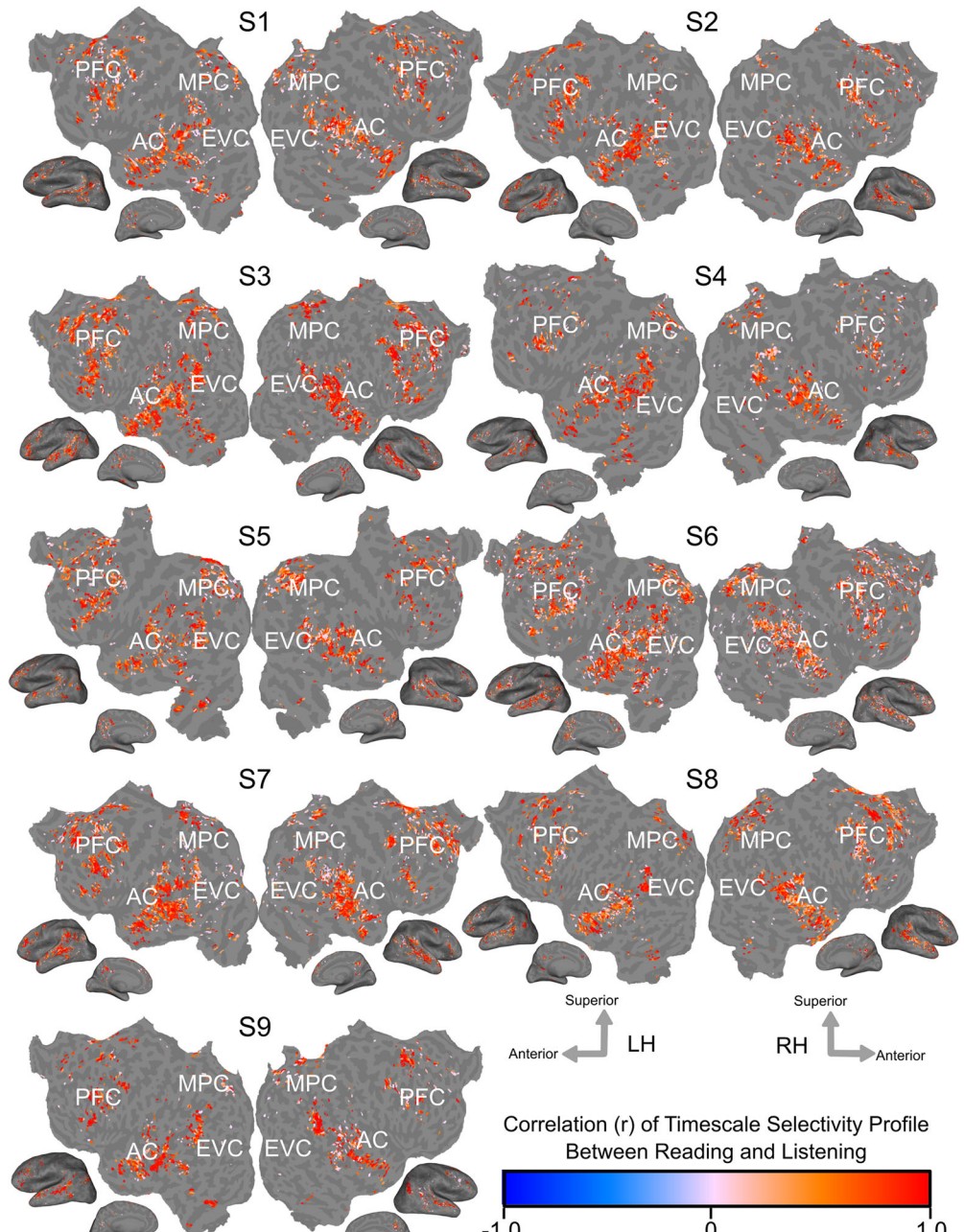

**Fig. 3 | Voxelwise similarity of timescale selectivity.** The Pearson correlation coefficient of the timescale selectivity profile between reading and listening is shown on the cortical surfaces of each participant. The correlation coefficient is shown according to the color scale at the bottom. Red voxels have positively correlated timescale selectivity profiles between reading and listening. Blue voxels have negatively correlated timescale selectivity profiles between reading and listening. Voxels that were not significantly predicted in both modalities are shown in gray (one-sided permutation test, $P < 0.05$, FDR-corrected). In areas that are language-selective in both modalities, the timescale selectivity profile is highly correlated across voxels.

individual participants (Fig. 1) and at the group level (Fig. 2). For both modalities, prefrontal and temporal cortices contain spatial gradients from intermediate to long-timescale selectivity, and precuneus is selective for long timescales. Comparisons of selectivity for each individual voxel (Fig. 3), and to each timescale separately (Fig. 4), show that the cortical representation of each timescale is similar between reading and listening. These results suggest that the topographic organization of language processing timescales is shared across stimulus modalities.

Prior work has studied brain representations of contextualized and non-contextualized language, separately for written[15] and spoken language comprehension[5]. Those studies showed that areas within the medial parietal cortex, prefrontal cortex, and inferior temporal cortex preferentially represent contextualized information; whereas other areas within superior temporal cortex and the temporoparietal junction do not show a preference for contextualized information. Our results build upon these previous findings by directly comparing representations between reading and listening within individual participants and by examining representations across a finer granularity of timescales. The fine-grained variation in timescale selectivity that we observed within previously proposed cortical networks supports the hypothesis that language processing occurs along a continuous gradient, rather than in distinct, functionally specialized brain networks[3].

Our study provides new evidence on the similarities in language processing between reading and listening. To compare brain responses between

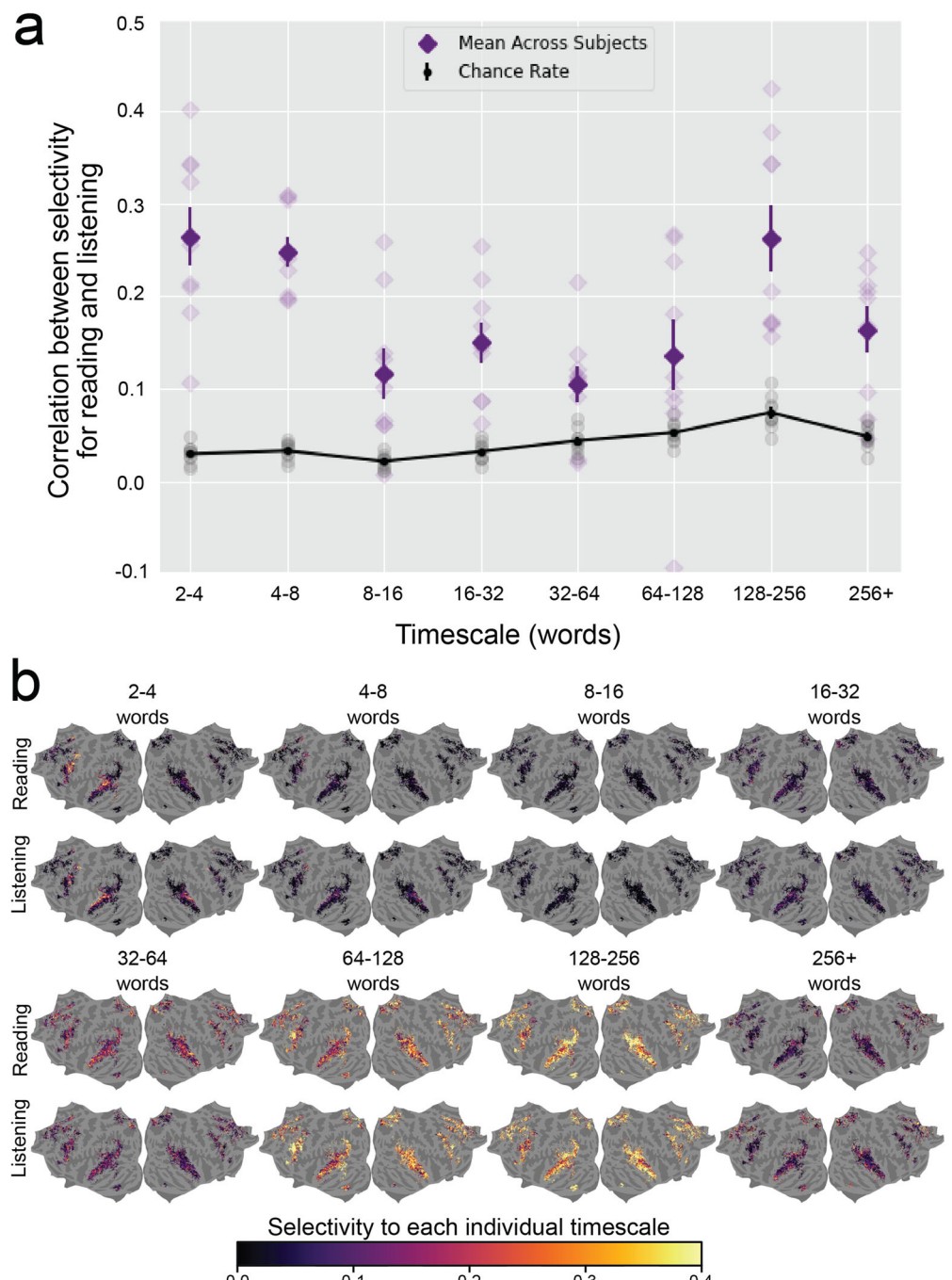

**Fig. 4 | Similarity of selectivity for each timescale between reading and listening.** Selectivity for each individual timescale was compared between reading and listening across the cerebral cortex. For each voxel, selectivity for each individual timescale describes the extent to which the corresponding timescale-specific feature space explains variation in BOLD responses, relative to the other timescale-specific feature spaces (see Methods for details). **a** For each timescale, the Pearson correlation coefficient was computed between selectivity for that timescale during reading and listening, across all voxels that were significantly predicted for both modalities. For each timescale, the mean true correlation across participants is indicated by dark purple diamonds. The mean chance correlation across participants is indicated by black dots (for clarity, these black dots are connected by a black line). Vertical lines through purple diamonds and through black dots are error bars that indicate the standard error of the mean (SEM) across participants for the respective value. True and chance correlations for each individual participant are respectively indicated by light purple diamonds and gray dots. The true correlation is significantly higher than chance in most individual participants and timescales; see Supplementary Table S1 for details. **b** The group-level selectivity of each vertex to each timescale is shown in fsAverage space for reading and listening separately. Vertices that were not language-selective in both modalities are shown in gray. Outside of primary sensory areas, selectivity for each timescale is distributed similarly across the cortical surface between both modalities. Among voxels that are language-selective in both modalities, each language timescale is represented in similar areas between reading and listening. These results further indicate that there is a shared organization of representations of language timescales between reading and listening.

reading and listening, prior work correlated timecourses of brain responses between participants who read and listened to the same stimuli[12]. That work found similarities in areas such as superior temporal gyrus, inferior frontal gyrus, and precuneus; and differences in early sensory areas as well as in parts of parietal and frontal cortices. However, that work did not specifically model linguistic features. Therefore the differences they observed between modalities in parietal and frontal cortices may indicate differences in non-linguistic processes, such as high-level control processes, rather than

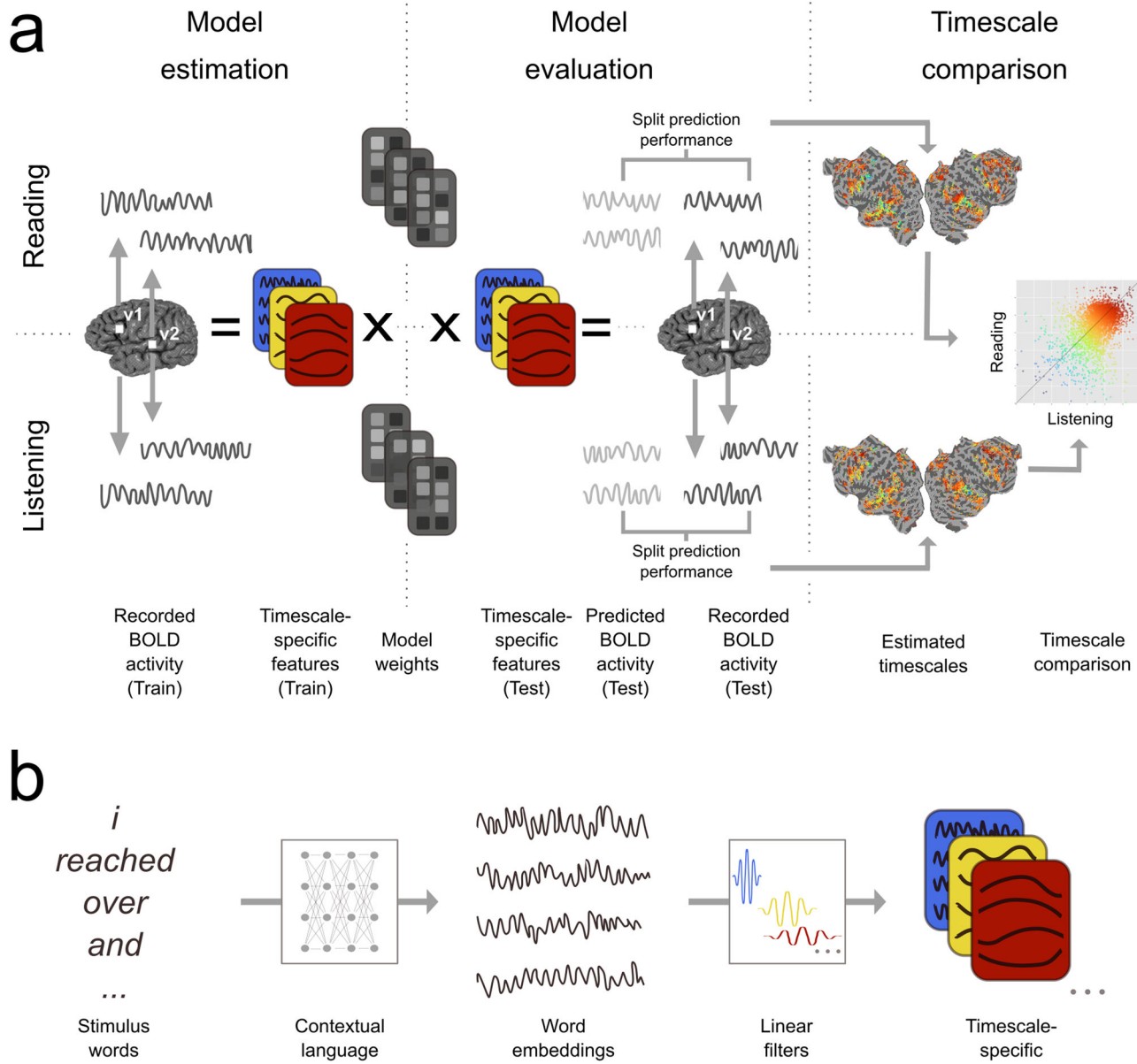

**Fig. 5 | Experimental procedure and voxelwise modeling.** The following procedure was used to compare the representation of different language timescales across the cerebral cortex. **a** Functional MRI signals were recorded while participants listened to or read narrative stories[13,16]. Timescale-specific feature spaces were constructed, each of which reflects the components of the stimulus that occur at a specific timescale (see (**b**) for details). These feature spaces and BOLD responses were used to estimate voxelwise-encoding models that indicate how different language timescales modulate the BOLD signal evoked in each voxel, separately for each participant and modality ("Model estimation"). Estimated model weights were used to predict BOLD responses to a separate held-out dataset which was not used for model estimation ("Model evaluation"). Predictions for individual participants were computed separately for listening and reading sessions. Prediction performance was quantified as the correlation between the predicted and recorded BOLD responses to the held-out test dataset. This prediction performance was used to determine the selectivity of each voxel to language structure at each timescale. These estimates were then compared between reading and listening ("Timescale comparison").
**b** Timescale-specific feature spaces were constructed from the presented stimuli. A contextual language model (BERT[35]) was used to construct a vector embedding of the stimulus. The resulting stimulus embedding was decomposed into components at specific timescales. To perform this decomposition, the stimulus embedding was convolved across time with each of eight linear filters. Each linear filter was designed to extract components of the stimulus embedding that vary with a specific period. This convolution procedure resulted in eight sets of stimulus embeddings, each of which reflects the components of the stimulus narrative that vary at a specific timescale. These eight sets of stimulus embeddings were used as timescale-specific feature spaces in (**a**).

differences in language representations. By specifically modeling representations of linguistic features, our results suggest that some of the differences observed in prior work[12] could indeed be due to non-linguistic processes such as high-level control. A separate study suggesting that brain representations of language differ between modalities compared brain responses to different types of stimuli for reading and listening: the stimuli used for reading experiments consisted of isolated sentences, whereas the stimuli used for listening experiments consisted of full narratives[18]. This discrepancy perhaps explains why in that study[18], language models trained on higher-level tasks (e.g., summarization, paraphrase detection) were better able to predict listening than reading data. Our study used matched stimuli for reading and listening experiments, and the similarities we observed highlight the importance of using narrative-length, naturalistic stimuli to elicit brain representations of high-level linguistic features[19].

The method for estimating timescale selectivity that we introduced in this work addresses limitations in methods previously used to study language timescales in the brain[2–6]. Early methods required the use of stimuli that are scrambled at different temporal granularities[2,3]. However, artificially scrambled stimuli may cause attentional shifts, evoking brain responses that are not representative of brain responses to natural stimuli[19–21]. Other approaches measured the rate of change in patterns of brain responses in order to determine the temporal granularity of representations in each brain region[4]. However, that approach does not provide an explicit stimulus-response model which is needed to determine whether the temporal granularity in each brain region reflects linguistic or non-linguistic brain representations. Our approach uses voxelwise modeling, which allows us to estimate brain representations with ecologically valid stimuli, and obtain an explicit stimulus-response model. Our method uses spectral analysis to extract stimulus features that reflect different language timescales, decoupling the feature extraction process from specific neural network architectures. This decoupling enables the construction of encoding models that are more accurate and that are also interpretable in terms of timescale selectivity. In the future, our method could be used with pretrained audio or visual models (e.g., wav2vec 2.0[22] or TrOCR[23]) to estimate selectivity for different timescales of low-level auditory and visual features. In sum, the method for estimating timescale selectivity that we developed in this study allowed us to produce more interpretable, accurate, and ecologically valid models of language timescales in the brain than previous methods.

To further inform theories of language integration in the brain, our approach of analyzing language timescales could be combined with approaches that analyze brain representations of specific classical language constructs. Approaches based on classical language constructs such as part-of-speech tags[24] and hierarchical syntactic constructs[25,26] provide intuitive interpretations of cortical representations. However, these language constructs do not encompass all the information that is conveyed in a natural language stimulus. For example, discourse structure and narrative processes are difficult to separate and define. This difficulty is particularly acute for freely produced stimuli, which do not have explicitly marked boundaries between sentences and paragraphs. Instead of classically defined language constructs, our approach uses spectral analysis to separate language timescales. The resulting models of brain responses can therefore take into account stimulus language information beyond language constructs that can be clearly separated and defined. In the future, evidence from these two approaches could be combined in order to improve our understanding of language processing in the brain. For example, previous studies suggested that hierarchical syntactic structure may be represented in the left temporal lobe, areas in which our analyses identified a spatial gradient from intermediate to long-timescale selectivity[26]. Evidence derived from both approaches should be further compared in order to inform neurolinguistic theories with a spatially and temporally fine-grained model of voxel representation that can be interpreted in terms of classical language constructs.

One limitation of our study comes from the temporal resolution of BOLD data. Because the data used in this study have a repetition time (TR) of 2 seconds, our analysis may be unable to detect very fine-grained distinctions in timescale selectivity. Furthermore, controlling for low-frequency voxel response drift required low-pass filtering of the BOLD data during preprocessing. This preprocessing filter may have removed information about brain representations of very long timescales (i.e., timescales above 360 words), thus removing information about these timescales. Future work could apply our method to brain recordings that have more fine-grained temporal resolution (e.g., from electrocorticography (ECoG) or electroencephalography (EEG) recordings) or that do not require low-pass filtering in order to determine whether there are subtle differences in timescale selectivity between modalities. A second limitation arises from the current state of language model embeddings. Although embeddings from language models explain a large proportion of variance in brain responses, these embeddings do not capture all stimulus features (e.g., features that change within single words). In the future, our method can be used with other language models to obtain more accurate estimates of timescale selectivity.

In sum, we developed a sensitive, data-driven method to determine whether language timescales are represented in the same way during reading and listening across cortical areas that represent both written and spoken language. Analyses of timescale selectivity in individual participants and at the group level reveal that the cortical representation of different language timescales is highly similar between reading and listening across temporal, parietal, and prefrontal cortices at the level of individual voxels. The shared organization of cortical language timescale selectivity suggests that a change in stimulus modality alone does not substantially alter the organization of representations of language timescales. A remaining open question is whether a change in the temporal constraints of language processing would alter the organization of representations of language timescales. One interesting direction for future work would be to compare whether a change in the stimulus presentation method (e.g., static text presentation compared to transient rapid serial visual presentation (RSVP)) would alter the organization of language timescale representations.

## Methods

Functional MRI was used to record BOLD responses while human participants read and listened to a set of English narrative stories[13,16]. The stimulus narratives were transformed into feature spaces that each reflect a particular set of language timescales. Each timescale was defined as the spectral components of the stimulus narrative that vary over a certain number of words. These timescale-specific feature spaces were then used to estimate voxelwise-encoding models that describe how different timescales of language are represented in the brain for each modality and participant separately. The voxelwise encoding models were used to determine the language timescale selectivity of each voxel, for each participant and modality separately. The language timescale selectivity of individual voxels was compared between reading and listening. The experimental procedure is summarized in Fig. 5 and is detailed in the following subsections.

### MRI data collection

MRI data were collected on a 3T Siemens TIM Trio scanner located at the UC Berkeley Brain Imaging Center. A 32-channel Siemens volume coil was used for data acquisition. Functional scans were collected using gradient echo EPI water excitation pulse sequence with the following parameters: repetition time (TR) 2.0045 s; echo time (TE) 31 ms; flip angle 70°; voxel size 2.24 × 2.24 × 4.1 mm (slice thickness 3.5 mm with 18% slice gap); matrix size 100 × 100; and field of view 224 × 224 mm. To cover the entire cortex, 30 axial slices were prescribed, and these were scanned in interleaved order. A custom-modified bipolar water excitation radiofrequency (RF) pulse was used to avoid signals from fat. Anatomical data were collected using a T1-weighted multi-echo MP-RAGE sequence on the same 3T scanner.

To minimize head motion during scanning and to optimize alignment across sessions, each participant wore a customized, 3D-printed or milled head case that matched precisely the shape of each participant's head[27,28]. In order to account for inter-run variability, within each run MRI data were z-scored across time for each voxel separately. The data presented here have been presented previously as part of other studies that examined questions unrelated to timescales in language processing[8,13,16]. Motion correction and automatic alignment were performed on the fMRI data using the FMRIB Linear Image Registration Tool (FLIRT) from FSL 5.0[29]. Low-frequency voxel response drift was removed from the data using a third-order Savitzky–Golay filter with a 120 second window (for data preprocessing details, see ref. 13).

### Participants

Functional data were collected on nine participants (six males and three females) between the ages of 24 and 36. All procedures were approved by the Committee for the Protection of Human Subjects at the University of California, Berkeley. All participants gave informed consent. All ethical regulations relevant to human research participants were followed. All

participants were healthy, had normal hearing, and had normal or corrected-to-normal vision. The Edinburgh handedness inventory[30] indicated that one participant was left-handed. The remaining eight participants were right-handed or ambidextrous.

Because the current study used a voxelwise-encoding model framework, each participant's data were analyzed individually, and both statistical significance and out-of-set prediction accuracy (i.e., generalization) are reported for each participant separately. Because each participant provides a complete replication of all hypothesis tests, sample size calculations were neither required nor performed.

## Stimuli

Human participants read and listened to a set of English narrative stories while in the fMRI scanner. The same stories were used as stimuli for reading and listening sessions, and the same stimuli were presented to all participants. These stories were originally presented at The Moth Radio Hour. In each story, a speaker tells an autobiographical story in front of a live audience. The selected stories cover a wide range of topics and are highly engaging. The stories were separated into a model training dataset and a model test dataset. The model training dataset consisted of ten 10–15-min stories. The model test dataset consisted of one 10-min story. This test story was presented twice in each modality (once during each scanning session). The responses to the test story were averaged within each modality (for details, see refs. 16 and 13). Each story was played during a separate fMRI scan. The length of each scan was tailored to the story and included 10 seconds of silence both before and after the story. The listening and reading presentation order was counterbalanced across participants.

During listening sessions, the stories were played over Sensimetrics S14 in-ear piezoelectric headphones. During reading sessions, the words of each story were presented one by one at the center of the screen using a rapid serial visual presentation (RSVP) procedure[10,31]. Each word was presented for a duration precisely equal to the duration of that word in the spoken story. The stories were shown on a projection screen at $13 \times 14°$ of visual angle. Participants were asked to fixate while reading the text (for details about the experimental stimuli, see ref. 13).

## Voxelwise-encoding models

Voxelwise modeling (VM) was used to model BOLD responses[8,13,16,32,33]. In the VM framework, stimulus and task parameters are nonlinearly transformed into sets of features (also called "feature spaces") that are hypothesized to be represented in brain responses. Linearized regression is used to estimate a separate model for each voxel. Each model predicts brain responses from each feature space (a model that predicts brain responses from stimulus features is referred to as an "encoding model"). The encoding model describes how each feature space is represented in the responses of each voxel. A held-out dataset that was not used for model estimation is then used to evaluate model prediction performance on new stimuli and to determine the significance of the model prediction performance.

## Construction of timescale-specific feature spaces

To operationalize the notion of language timescales, the language stimulus was treated as a time series and different language timescales were defined as the different frequency components of this time series. Although this operational definition is not explicitly formulated in terms of classic language abstractions such as sentences or narrative chains, the resulting components nonetheless selectively capture information corresponding to the broad timescales of words, sentences, and discourses[34]. To construct timescale-specific feature spaces, first an artificial neural language model ("BERT"[35]) was used to project the stimulus words onto a contextual word embedding space. This projection formed a stimulus embedding that reflects the language content in the stimuli. Then, linear filters were convolved with the stimulus embedding to extract components that each vary at specific timescales. These two steps are detailed in the following two paragraphs.

## Embedding extraction

An artificial neural network (BERT-BASE-UNCASED[35]) was used to construct the initial stimulus embedding. BERT-base is a contextual language model that contains a 768-unit embedding layer and 12 transformer layers, each with a 768-unit hidden state (for additional details about the BERT-base model see ref. 35). The $w$ words of each stimulus narrative $X$ were tokenized and then provided one sentence at a time as input to the pretrained BERT-base model (sentence-split inputs were chosen as input context because sentence-level splits mimic the inputs provided to BERT during pretraining). For each stimulus word, the activation of each of the $p = 13 \times 768 = 9984$ units of BERT was used as a $p$-dimensional embedding of that word. Prior work suggested that language structures with different timescales are preferentially represented in different layers of BERT[36–38] (though some have argued that language timescales are not cleanly separated across different layers of BERT[39]). Earlier layers represent lower-level, shorter-timescale information (e.g., word identity and linear word order), whereas later layers represent higher-level, longer-timescale information (e.g., coreference, long-distance dependencies). To include stimulus information at all levels of the language processing hierarchy, activations from all layers of BERT were included in the stimulus embedding. The embeddings of the $w$ stimulus words form a $p \times w$ stimulus embedding $M(X)$. $M(X)$ numerically represents the language content of the stimulus narratives.

## Timescale separation

The stimulus embedding derived directly from BERT can explain a large proportion of the variance in brain responses to language stimuli[15,40–42]. However, this stimulus embedding does not distinguish between different language timescales.

In order to distinguish between different language timescales, linear filters were used to decompose the stimulus embedding $M(X)$ into different language timescales. Intuitively, the stimulus embedding consists of components that vary with different periods. Components that vary with different periods can be interpreted in terms of different classical language structures[34]. For example, components that vary with a short period (~2–4 words) reflect clause-level structures such as syntactic complements, components that vary with an intermediate period (~16–32 words) reflect sentence-level structures such as constituency parses, and components that vary with a long period (~128–256 words) reflect paragraph-level structures such as semantic focus. To reflect this intuition, different language timescales were operationalized as the components of $M(X)$ with periods that fall within different ranges. The period ranges were chosen to be small enough to model timescale selectivity at a fine-grained temporal granularity, and large enough to avoid substantial spectral leakage which would contaminate the output of each filter with components outside the specified timescale. The predefined ranges were chosen as: 2–4 words, 4–8 words, 8–16 words, 16–32 words, 32–64 words, 64–128 words, 128–256 words, and 256+ words. To decompose the stimulus embedding into components that fall within these period ranges, eight linear filters $b_i$ ($i \in 1, 2, \ldots, 8$) were constructed. Each filter $b_i$ was designed to extract components that vary with a period in the predefined range. The window method for filter design was used to construct each filter[43]. Each linear filter was constructed by multiplying a cosine wave with a blackman window[44]. The stimulus embedding $M(X)$ was convolved with each of the eight filters separately to produce eight filtered embeddings $M_i(X)$, $i \in 1, 2, \ldots, 8$, each with dimension $p \times w$. To avoid filter distortions at the beginning and end of the stimulus, a mirrored version of $M(X)$ was concatenated to the beginning and end of $M(X)$ before the filters were applied to $M(X)$. Each filtered embedding $M_i(X)$ contains the components of the stimulus embedding that vary at the timescale extracted by the $i$th filter.

## Construction of sensory-level feature spaces

Two sensory-level feature spaces were constructed in order to account for the effect of low-level sensory information on BOLD responses. One feature space represents low-level visual information. This feature space was

constructed using a spatiotemporal Gabor pyramid that reflects the spatial and motion frequencies of the visual stimulus (for details, see refs. [13],[45], and [14]). The second feature space represents low-level auditory information. This feature space was constructed using a cochleogram model that reflects the spectral frequencies of the auditory stimulus (for details, see refs. [8],[13], and [14]).

## Stimulus downsampling

Feature spaces were downsampled in order to match the sampling rate of the fMRI recordings. The eight filtered timescale-specific embeddings $M_i(X)$ contain one sample for each word. Because word presentation rate of the stimuli is not uniform, directly downsampling the timescale-specific embeddings $M_i(X)$ would conflate long-timescale embeddings with the presentation word rate of the stimulus narratives[6]. To avoid this problem, a Gaussian radial basis function (RBF) kernel was used to interpolate $M_i(X)$ in order to form intermediate signals $M'_i(X)$, following ref. [6]. Each $M'_i(X)$ has a constant sampling rate of 25 samples per repetition time (TR). After this interpolation step, an anti-aliasing, 3-lobe Lanczos filter with cut-off frequency set to the fMRI Nyquist rate (0.25 Hz) was used to resample the intermediate signals $M'_i(X)$ to the middle time points of each of the $n$ fMRI volumes. This procedure produced eight timescale-specific feature spaces $F_i(X)$, each of dimension $p \times n$. Each of these feature spaces contains the components of the stimulus embedding that vary at a specific timescale. These feature spaces are sampled at the sampling rate of the fMRI recordings. The sensory-level feature spaces were not sampled at the word presentation rate. Therefore, Gaussian RBF interpolation was not applied to sensory-level feature spaces.

Before voxelwise modeling, each stimulus feature was truncated, z-scored, and delayed. Data for the first 10 TRs and the last 10 TRs of each scan were truncated to account for the 10 s of silence at the beginning and end of each scan and to account for non-stationarity in brain responses at the beginning and end of each scan. Then the stimulus features were each z-scored in order to account for z-scoring performed on the MRI data (for details, see "MRI data collection"). In the z-scoring procedure, the value of each feature channel was separately normalized by subtracting the mean value of the feature channel across time and then dividing by the standard deviation of the feature channel across time. Note that the resulting feature spaces had low correlation with each other—for each pair of feature spaces, the mean pairwise correlation coefficient between dimensions of the feature spaces was less than 0.1. Lastly, finite impulse response (FIR) temporal filters were used to delay the features in order to model the hemodynamic response function of each voxel. The FIR filters were implemented by concatenating feature vectors that had been delayed by 2, 4, 6, and 8 seconds[13],[14],[16].

## Voxelwise-encoding model fitting

Voxelwise-encoding models were estimated in order to determine which features are represented in each voxel. Each model consists of a set of regression weights that describes BOLD responses in a single voxel as a linear combination of the features in a particular feature space. In order to account for potential complementarity between feature spaces, the models were jointly estimated for all ten feature spaces: the eight timescale-specific feature spaces, and the two sensory-level feature spaces (the two sensory-level feature spaces reflect spectrotemporal features of the auditory stimulus and motion energy features of the visual stimulus)[46],[47].

Regression weights were estimated using banded ridge regression[46]. Unlike standard ridge regression, which assigns the same regularization parameter to all feature spaces, banded ridge regression assigns a separate regularization hyperparameter to each feature space. Banded ridge regression thereby avoids biases in estimated model weights that could otherwise be caused by differences in feature space distributions. Mathematically, the $m$ delayed feature spaces $F_i(X)$, $i \in 1, 2, \ldots, m$ (each of dimension $p$) were concatenated to form a feature matrix $F'(X)$ (dimension $(mp) \times n$). Then banded ridge regression was used to estimate a mapping $B$ (dimension $v \times (\sum_{i=1}^{m} p)$) from $F'(X)$ to the matrix of voxel responses $Y$ (dimension $v \times n$). $B$ is estimated according to

$\hat{B} = \arg\min_B ||Y - BF'(X)||_2^2 + \lambda||CB||_2^2$. A separate regularization parameter was fit for each voxel, feature space, and FIR delay. The diagonal matrix $C$ of regularization hyperparameters for each feature space and each voxel is optimized over tenfold cross-validation.

## Regularization hyperparameter selection

Data for the ten narratives in the training dataset were used to select regularization hyperparameters for banded ridge regression. Tenfold cross-validation was used to find the optimal regularization hyperparameters for each feature space and each voxel. Regularization hyperparameters were chosen separately for each participant and modality. In each fold, data for nine of the ten narratives were used to estimate an encoding model and the tenth narrative was used to validate the model. The regularization hyperparameters for each feature space and voxel were selected as the hyperparameters that produced the minimum squared error (L2) loss between the predicted voxel responses and the recorded voxel responses $(\arg\min_{hyperparameters} ||\hat{y} - y||_2^2)$. Because evaluating $k$ regularization hyperparameters for $m$ feature spaces requires $k^m$ iterations $(10^{10} = 10,000,000,000$ model fits in our case), it would be impractical to conduct a grid search over all possible combinations of hyperparameters. Instead, a computationally efficient two-stage procedure was used to search for hyperparameters[47]. The first stage consisted of 1000 iterations of a random hyperparameter search procedure[48]. In total, 1000 normalized hyperparameter candidates were sampled from a dirichlet distribution and were then scaled by 10 log-spaced values ranging from $10^{-5}$ to $10^5$. Then the voxels with the lowest 20% of the cross-validated L2 loss were selected for refinement in the second stage. The second stage consisted of 1000 iterations of hyperparameter gradient descent[49]. This stage was used to refine the hyperparameters selected during the random search stage. This hyperparameter search was performed using the Himalaya Python package[47]. Note that hyperparameter selection in banded ridge regression acts as a feature-selection mechanism that helps account for stimulus feature correlations[47].

## Model estimation and evaluation

The selected regularization hyperparameters were used to estimate regression weights that map from the timescale-specific feature spaces to voxel BOLD responses. Regression weights were estimated separately for each voxel in each modality and participant. The test dataset was not used to select hyperparameters or to estimate regression weights. The joint prediction performance $r$ of the combined feature spaces was computed per voxel as the Pearson correlation coefficient between the predicted voxel responses and the recorded voxel responses. The split-prediction performance $\tilde{r}$ was used to determine how much each feature space contributed to the joint prediction performance $r$. The split-prediction performance decomposes the joint prediction performance $r$ of all the feature spaces into the contribution $\tilde{r}_i$, $i \in 1, 2, \ldots m$ of each feature space. The split-prediction performance is computed as $\tilde{r}_i = \frac{\sum_t \hat{Y}_i[t] Y[t]}{\sqrt{\left(\sum_t \hat{Y}[t]^2\right)\left(\sum_t Y[t]^2\right)}}$, where $t$ denotes each timepoint (further discussion of this metric can be found in refs. [50] and [47]).

## Language-selective voxel identification

The set of "language-selective voxels" was operationally defined as the set of voxels that are accurately predicted by any of the eight timescale-specific feature spaces. To identify this set of voxels, the split-prediction performance was used. The total contribution $\tilde{r}_{all\_timescales}$ of the eight timescale-specific feature spaces to predicting the BOLD responses in each voxel was computed as the sum of the split-prediction performance for each of the eight timescales $\tilde{r}_{all\_timescales} = \sum_{i=1}^{8} \tilde{r}_{timescale_i}$. The significance of $\tilde{r}_{all\_timescales}$ was computed by a permutation test with 1000 iterations. At each permutation iteration, the timecourse of the held-out test dataset was permuted by blockwise shuffling (shuffling was performed in blocks of 10 TRs in order to

account for autocorrelations in voxel responses[6,13]). The permuted timecourse of voxel responses was used to produce a null estimate of $\tilde{r}_{all\_timescales}$. These permutation iterations produced an empirical distribution of 1000 null estimates of $\tilde{r}_{all\_timescales}$ for each voxel. This distribution of null values was used to obtain the $P$ value of $\tilde{r}_{all\_timescales}$ for each voxel separately. A false discovery rate (FDR) procedure was used to correct the resulting $P$ values for multiple comparisons within each participant and modality[51]. A low $P$ value indicates that the timescale-specific feature spaces significantly contributed to accurate predictions of BOLD responses in the joint model. Voxels with a one-sided FDR-corrected $P$ value of less than $P < 0.05$ were identified as language-selective voxels. The set of language-selective voxels was identified separately for each participant and modality.

### Voxel timescale selectivity estimation

The encoding model estimated for each voxel was used to determine voxel timescale selectivity, which reflects the average language timescale for which a voxel is selective. In order to compute timescale selectivity, first the timescale selectivity profile ($\tilde{r}'$) was computed. The timescale selectivity profile reflects the selectivity of each voxel to each of the eight timescale-specific feature spaces. This metric is computed by normalizing the vector of split-prediction performances of the eight timescale-specific feature spaces to form a proper set of proportions: $\tilde{r}'_{timescale_i} = \frac{max(0, \tilde{r}_{timescale_i})}{\sum_{j=1}^{8} max(0, \tilde{r}_{timescale_j})}$.

Comparing each index of the timescale selectivity profile separately cannot distinguish between cases in which a voxel represents similar timescales between reading and listening (e.g., 2–4 words for reading and 4–8 words for listening) and cases in which a voxel represents very different timescales between the two modalities (e.g., 2–4 words for reading and 128–256 words for listening). Therefore, we computed the timescale selectivity $\bar{T}$ for each voxel, which reflects the average timescale of language to which a voxel is selective (we use the weighted average instead of simply taking the maximum selectivity across timescales, in order to prevent small changes in prediction accuracy from producing large changes in estimated timescale selectivity). To compute voxel timescale selectivity, first the timescale $t_i$ of each feature space $F_i(X)$ was defined as the center of the period range of the respective filter $b_i$: $t_i = \frac{p_{i,low} + p_{i,high}}{2}$, where $(p_{i,low}, p_{i,high})$ indicates the upper and lower end of the period range for filter $i$. Then, timescale selectivity was defined as a weighted sum of each feature space log-timescale: $\bar{T} = 2^{\left(\sum_{i=1}^{8}(\tilde{r}'_i \log_2(t_i))\right)}$. Timescale selectivity was computed separately for each voxel, participant, and modality.

### Voxel timescale comparison

To compare timescale selectivity between modalities, the Pearson correlation coefficient was computed between timescale selectivity during reading and listening across the set of voxels that are language-selective in both modalities. The significance of this correlation was determined by a permutation test with 1000 iterations. At each iteration and for each modality separately, the timecourse of recorded voxel responses was shuffled. The timecourses were shuffled in blocks of 10 TRs in order to account for autocorrelations in voxel responses. The shuffled timecourses of recorded voxel responses were used to compute a null value for the timescale selectivity of each voxel for each modality separately. The null values of timescale selectivity were correlated between reading and listening to form an empirical null distribution. This null distribution was used to determine the $P$ value of the observed correlation between timescale selectivity during reading and listening. Significance was computed for each participant separately.

In addition, for each of the eight timescales separately, the Pearson correlation coefficient was computed between selectivity for that timescale during reading and listening. This correlation was performed across the set of voxels that are language-selective in both modalities. The significance of the observed correlations was computed by a permutation test. At each of 1000 iterations, the timecourse of recorded voxel responses was shuffled, and then the shuffled voxel responses were used to compute null values of

the timescale selectivity profile. For each timescale-specific feature space separately, the null values of the timescale selectivity profile were used to compute an empirical null distribution for the correlation between selectivity for that feature space during reading and listening. These null distributions were used to determine the $P$ value of the observed correlations. Significance was computed for each participant and for each timescale-specific feature space separately.

### Statistics and reproducibility

Data were analyzed for each of the nine participants separately. Statistical significance and out-of-set prediction accuracy (i.e., generalization) are reported for each participant separately. Results were reproduced across all participants.

### Reporting summary

Further information on research design is available in the Nature Portfolio Reporting Summary linked to this article.

### Data availability

This study made use of data originally collected for separate studies[8,13,16]. The data can be accessed at https://gin.g-node.org/denizenslab/narratives_reading_listening_fmri/. The source data for figures in this paper is available at https://gin.g-node.org/denizenslab/narratives_reading_listening_fmri/src/master/chen2024_timescales.

### Code availability

Custom code for this study is available at https://github.com/denizenslab/timescales_filtering. All model fitting and analysis were performed using custom software written in Python, making heavy use of NumPy[52], SciPy[53], Matplotlib[54], Himalaya[47], and Pycortex[55]. The BERT-BASE-UNCASED model was accessed via Huggingface[56].

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

## Acknowledgements
C.C. was supported by the National Science Foundation (Nat-1912373 and DGE 1752814) and an IBM PhD Fellowship. F.D. was supported by the Federal Ministry of Education and Research (BMBF 01GQ1906) and the Berliner ChancengleichheitsProgramm (BCP). Data collection for this work was additionally funded by the National Science Foundation (IIS1208203) and the National Eye Institute (EY019684 and EY022454).

## Author contributions
C.C. and F.D. conceptualized the experiment. F.D. collected the data. C.C., T.D.T., and F.D. developed the methodology and contributed to the analysis. C.C., T.D.T., J.L.G., D.K., and F.D. wrote the paper.

## Competing interests
The authors declare no competing interests.
