## [Peer Review File · Communications Biology]

Reviewers' comments:

Reviewer #1 (Remarks to the Author):

In this manuscript, Chen et al., used voxelwise encoding models to investigate the topographic organization of language timescale representations across the cortical surface during both written and spoken language comprehension. The authors convincingly showed voxel selectivity to increasingly complex language processing timescales to form spatial gradients across prefrontal and temporal cortex, while also showing this topographic organization to be preserved across stimulus modality. The authors' analytic approach impressively corroborates and adds to earlier work on the subject while simultaneously correcting for previous limitations concerning timescale precision and ecological validity. However, we do have critiques concerning their use of BERT, interpretation of findings, and inconsistencies in figures.

Major Points:

Stimulus embedding:

Better justification is needed for the approach the authors took to constructing their stimulus embedding. In concatenating the activations of all 13 layers of BERT to get a p-dimensional ($p = 13 \times 768$) embedding per word, the authors cite three articles which supposedly point to this method as producing a "stimulus embedding [that] can explain a large proportion of the variance in brain responses to language stimuli." However, following closer inspection of the articles referenced as justification, they each seem to suggest that the use of embeddings derived from intermediate layers produces models which are most predictive of brain activity during spoken language comprehension, consistently outperforming models which use representations from the most peripheral layers. Indeed, it appears each study evaluates stimulus embeddings on a layer-by-layer basis (sometimes ignoring the input representation), ultimately reporting the brain predictivity of the model which used the best-performing layer. Nowhere, it seems, do they explicitly state the use of a stimulus embedding which concatenates the activations of all layers in tandem. Indeed, a surface reading suggests that incorporating the outermost layers into the stimulus embedding may confound the performance of their encoding models.

Still, this doesn't necessarily mean that including the activations from all of BERT's layers produces a poor stimulus embedding for their purposes; the authors just need to better justify why they took this approach. Because the first layer BERT receives is simply the sum of the token, segmentation, and position embeddings, the lower layers primarily represent linear word order, with linear/sequential information beginning to drop off at about layer 4. As such, as you advance up BERT's layers, BERT phases out positional information in favor of syntactically-sensitive hierarchical information, with syntactic information being most prominent in the representations of BERT's middle layers. Relative to syntactic information, semantic information is less localized to a given subset of layers but is best represented in higher layers. Indeed, the representations of deeper layers are required for BERT to handle difficult cases of subject-verb agreement that rely on long-range dependencies. The deepest

layers of BERT are the most task specific. In the absence of fine-tuning, the activations of the final layer of the pre-trained BERT-based-uncased model the authors used are more specific to the masked language modeling (MLM) and next sentence prediction (NSP) tasks the model was trained on relative to earlier layers. All in all, BERT is said to have “rediscovered the classical NLP pipeline” as BERT, by and large, encodes basic syntactic features at the bottom-most layers (e.g., part-of-speech), high-level syntactic information in the middle layers (e.g., constituents), and complex semantic information at higher layers (e.g., semantic role labeling, coreference). Therefore, considering the authors are investigating the topographic representation of language structures of increasing complexity (e.g., “clause-level structures” → “paragraph-level structures”), perhaps concatenating the activations of all of BERT’s layers can be justified as being an ‘all-inclusive approach’ that does not favor layer representations biased to a given timescale. Indeed, we suspect a reason why the three papers the authors cited found the intermediate layer representations of transformers to be best at predicting brain activity has to do with the fact that the middle layers of transformer-based language models tend to be the most transferable across a diverse range of NLP ‘probing’ tasks; i.e., the representations of intermediate layers tend to generalize the most across language timescales and therefore may be more predictive of voxel activity on average.

Nevertheless, work which seeks to identify what kinds of linguistic information are represented most robustly and transparently in each of (and across) BERT’s layers is ongoing and does not suggest that knowledge of different language timescales is equally captured in the full model representation. Their use of linear filters to acquire timescale-specific features is intriguing and appears to have produced robust results, but it is not obvious to what extent the pre-filtered contextualized embeddings themselves differentially represent information of different language timescales.

Ultimately, we believe the approach the authors took to constructing their stimulus embedding may be reasonable but needs to be better justified, including a discussion of possible confounds in their approach. Additionally, while we do not presume that their chief claim concerning the shared topographic organization of language timescale representations between written and spoken language comprehension will need modification, exploring how this conclusion holds when experimenting with different stimulus embeddings (e.g., using just the intermediate BERT layers) seems to us to be a necessary control.

RSVP reading:

We understand that to make reading and listening more comparable, each word during the reading session was presented for a duration equal to the rate of that word’s presentation in the listening session. While matching the timing of word presentation across stimulus modalities is a good control, we worry that RSVP reading may be more similar to natural listening than natural reading, which would of course confound the claim that the cortical representation of language timescales is shared between reading and listening. Not only is RSVP reading not self-paced, but given each word is presented individually, participants are forced to read linearly and cannot look to neighboring words at any given time. We speculate that reading in this fashion may incline participants to rely on an internal monologue, verbalizing each word in their mind, more so than they would during natural reading. That is, participants may be ‘listening’ to their own inner voice more than they normally would during natural

reading and thus be activating auditory cortex-based mechanisms more than during natural reading. Conclusions about shared mechanisms across modalities may then be over-stated (including in the title) and need to be appropriately qualified.

Within- vs. between-modality differences:

The authors do a good job demonstrating, at both the individual and group level, that there exists a topographic organization of language processing timescales that is shared between reading and listening. While the paper's focus is of course on cross-modal similarities, we believe that the authors do not sufficiently emphasize the differences in timescale representation between the stimulus modalities. Several figures appear to show differences between modalities in short timescale representations in unimodal sensory cortices. However, these differences are never explicitly discussed or measured in the results. Also, when the authors claim that timescale selectivity is significantly positively correlated between reading and listening, they only consider voxels that are language-selective to both modalities. As a consequence, in both the discussion and statistical analysis, modality-invariant language representations appear to be overemphasized, understating the differences in timescale selectivity between reading and listening and perhaps statistically overinflating the extent of similarity. Doing so may even serve to support the ecological validity of their RSVP reading paradigm by showing how the reading paradigm does indeed engage different areas from the listening paradigm.

Sequence input context length:

It may be worthwhile for the authors to provide justification as to why they provided one sentence at a time as input, when BERT-base models handle a maximum input length of 512 tokens. Considering a goal is to extract timescale features related to long-range dependencies between distant tokens, it isn't immediately obvious to us as to why the authors wouldn't extend their context length window to be longer, or experiment with different input sequence lengths, instead of settling on word embeddings which are only locally contextualized. In our quick reading of the Jain and Huth (2018) paper that's referenced in the manuscript, an appropriate justification may be that voxelwise encoding model performance increases monotonically as a function of context length, hitting a point of diminishing returns around the length of an average sentence. However, the J&H paper only examines this relationship up to a context length of 20 and uses a LSTM language model. Nevertheless, the Toneva and Wehbe (2019) paper the authors cited uses BERT and found that encoding model performance (voxel prediction accuracy) improves monotonically up to about 15 words (depending on the layer) and drops off after as context length increases (at least up to 40 words). Therefore, feeding BERT one sentence at a time may be reasonable, especially when considering that the complexity of BERT's attention mechanism is quadratic with respect to context length; however, the authors need to make their reasoning explicit.

Be that as it may, the J&H (2018) study found that language timescale selectivity can be probed, albeit in a rather confounded and uninterpretable way, by varying context length. They found that low-level language areas (e.g., auditory cortex) are selective to contextual LSTM representations built from short sequence inputs, whereas higher-language areas prefer LSTM embeddings built from inputs of longer lengths which incorporate more context. It is unclear to us whether the authors' method of extracting timescale-specific features is confounded by using a rather short context length, especially since

sentences are themselves relatively self-contained units of meaning whereas J&H (2018) appear to treat their input sequences as an arbitrary span of contiguous text; however, we believe it is worth investigating.

Ultimately, we believe it necessary that the authors evaluate how their conclusions hold when systematically varying the context length (i.e., number of tokens) passed to BERT-base for two reasons: (1) timescale representations of language in the brain have been shown to be sensitive to the amount of context given to language models, and (2) it is unclear whether the longer-range timescale features fail to adequately represent long-range information as they are extracted from stimulus embeddings that are only contextualized in the short to intermediate range. We anticipate that varying context length will not result in the primary claims of the manuscript needing to be significantly modified, but firmly believe that this still needs to be explicitly tested given how the authors' methods differ from previous studies.

Minor Points:

Mismatch in timescale selectivity patterns between the flattened cortical surface and inflated cortex: In Figure 2B, the activity patterns shown on the flattened versus inflated cortical representations do not appear to match. The inflated representations have a lot of mid-level selective voxels (green/blue) in lateral occipital cortex whereas the flattened ones do not. This appears to be because the authors portrayed vertices that were significantly predicted for both modalities (in at least 1/3 of the participants) on the flattened cortical surface, but instead included vertices specifically predicted within-modality on the inflated cortex. This figure and the others should be checked for accuracy.

Map timescale selectivity patterns against known networks:

Given that the data have been projected to a cortical surface, the authors could easily take the extra step to plot the voxelwise results against known brain networks. In particular, showing whether the long-timescale voxels map onto the default mode and cognitive control networks would help contextualize the findings in terms of other studies that have looked at, for instance, how the default mode network is involved in theory-of-mind operations. In addition, some of the transitions between short and long-timescale representations look more categorical than gradual and may correspond to known borders separating auditory, language, and default-mode networks. Similarly, plotting the results against language atlases (i.e., the ones available from the Fedorenko lab) will help put these results in the context of the known language network architecture. Doing this will not only improve readability but also reliability, as other researchers' results can be compared in the same anatomical framework.

Reviewer #2 (Remarks to the Author):

This work investigates the timescales of language representations during listening and reading of the same naturalistic story. The authors use voxelwise encoding models to predict fMRI recordings as a

function of timescale representations that are derived from a spectral decomposition of contextualized word embeddings from a large language model (BERT). The main claim in the work is that the timescales of language representations during reading and listening are the same, based on results showing significant correlations between the timescale selectivity during listening vs reading for a large number of voxels that are typically associated with language comprehension.

In general, I believe the approach proposed by the authors has promise. However, I am not yet convinced of their claim because of concerns relating to their methodology and analysis choices.

Methodology questions:

1. Are the features at different timescales correlated with one another? Such correlations would impede disentangling these features's contributions to the brain prediction. How does the split-prediction performance method that the authors use deal with such correlations?
2. Doesn't the current formulation of timescale selectivity equate cases where there is very peaked selectivity for one frequency band, and cases where there is more uniform selectivity across frequency bands? I would think that we want to have a way to distinguish those cases.
3. It's not clear how the contextualized word representations were extracted from BERT. Was a masked token used? Was the representation at each word in the input sentence extracted, or was it extracted only at the last token? What was done with words that have multiple tokens? BERT is a bidirectional model, so if the representations of all words were extracted instead of just at the last word, then information about future words is included in many of the word representations. Can the authors comment on this choice and how they think it affects their results?

Additional analyses that would strengthen the claims:

4. How do the results align with those from Lerner et al. 2011 that suggest that several language regions are sensitive to short and long timescales? In general, I would like to better understand the estimated selectivity across timescales – is it mostly peaked at one timescale, or is uniform across several close timescales, and is there some systematic shape in the distribution of selectivity across regions? From the Lerner et al. 2011 results, I would expect that the STS would have a more uniform distribution of selectivity, whereas the PC would have a peaked distribution at the long timescales.
5. Re Fig 3 results: it's strange to compute correlations across all voxels for reading vs listening to indicate selectivity because the authors already showed in Fig 2 that there is a large difference across voxels! I believe a much more straightforward analysis would be to correlate the vectors of timescale selectivity for listening vs reading, for each voxel. That would yield a voxel-wise estimate of how similar the modality-specific selectivity profiles are.
6. Do the authors believe the representations they construct capture all relevant language timescales for both modalities? For example, it doesn't seem that many voxels prefer short timescales currently, but is it possible that BERT does not capture timescales that are short enough because it is a text-only model? Would the selectivity results of listening vs reading be different given representations from an audio model, such as wav2vec, rather than a text-only model?

Better discussion of related work:

7. The authors claim that “prior work has not studied the organization of timescale representations during written language comprehension”. Some related work that should be discussed here is that of Toneva and Wehbe 2019, which is already referenced by the authors, but should also be further discussed in light of this claim. This previous work shows that, during written language comprehension, brain regions that are thought to be sensitive to long timescales (e.g. AG, PC, dmPFC; Lerner et al. 2011) are better predicted by contextualized word embeddings with longer contexts than by non-contextualized word embeddings. In contrast, regions that are thought to be sensitive to both short and long timescales (e.g. STS, IFG) are predicted equally well by non-contextualized as well as contextualized embeddings.

8. Additionally, Oota et al. 2022 <https://aclanthology.org/2022.naacl-main.235.pdf> investigate the voxelwise encoding performance during both listening and reading, and find differences in which task-optimized models predict each modality well. One could argue that the different tasks that these models are optimized for may relate to different timescales of language. Oota and colleagues use two datasets corresponding to different stimuli so it’s harder to make direct claims about differences between reading and listening, but the authors should at least discuss this previous work since their results can be seen to disagree with these previous findings.

Better explaining or toning down some statements:

9. L102: “These group-level results indicate that across participants, representations of language timescales are organized in the same way between reading and listening.” This statement is too strong, given both the methods and the results. What the results show is that the timescales *that are captured in the constructed representations* *align significantly* between the two modalities. There can be other timescales that are not captured by the current representations that do not align between the two modalities.

10. L39: However, it is unclear whether within these cortical areas written and spoken language elicit the same or different topographic organizations of language timescale representations. The answer to this question would suggest whether the same pathways are used to integrate written and spoken language. I don’t quite follow the logical implications here. Why would similar topographic organizations of language timescale suggest that the same pathways are being used to integrate written and spoken language? Perhaps it is slightly more granular information than just showing that the same regions are involved in language processing from both modalities, but I fail to see how it’s conclusive evidence that the same pathways are involved.

Minor:

The writing is for the most part clear and concise. However, there needs to be some description of what the timescale selectivity representation is in the main text. Currently, it is very difficult to understand the results without actually reading the methods pretty thoroughly.

L 192: the reference should appear in parentheses

Reviewer #3 (Remarks to the Author):

In this study the authors make use of an fMRI dataset in which subjects read and listened to narratives. They propose a novel method for building a mapping from words to neural responses (using a pre-trained Natural Language Processing transformer model) and then decomposing this mapping into different timescales. This allows them to identify the timescale of stimulus information that each voxel is sensitive to, without requiring experimental manipulations like scrambling stimulus content. They find that there is a clear hierarchy of response timescales throughout the brain, and that this timescale is largely similar between reading and listening.

This is a novel and elegant approach to identifying timescales of responses to language, which connects to but also extends previous work on naturalistic response timescales. The figures are simple but compelling, with patterns even in individual subjects. The finding that this timescale hierarchy is consistent across subjects *and* across modalities is very impressive, and provides an "atlas" that could be used in many future studies. This work should be of great interest to multiple subfields of cognitive neuroscience, including language perception, naturalistic dynamics, and neuroimaging methods.

My comments are primarily about the details of the novel analysis being proposed, which could use some additional clarification and justification.

Major comments:

1) The specific way that the word embeddings are being obtained involves a few analysis choices which could be better justified. Specifically:

a) What is the motivation for using BERT specifically, rather than other language models? Models trained on other tasks could potentially provide better (or different) feature embeddings. For example, cosine distances between the top-level embeddings from BERT have been shown to perform poorly on semantic similarity tasks (Reimers & Gurevych, 2019, arXiv:1908.10084), though that concern may not apply here, since regression weights are being learned on top of the embeddings.

b) The authors compute word embeddings by putting one sentence at a time into BERT. Does this bias the features toward a sentence-level timescale? E.g. does it limit the extent to which paragraph-level dependencies could appear in the feature space?

c) One unusual property of computing word embeddings in a bidirectional language model is that the feature representation of a word can depend on words that come after it in the sentence. Does this make the feature representations at fast timescales (2-4 words, 4-8 words) difficult to interpret?

2) Similarity, I'd like to see more justification for the choice of how the features were decomposed into separate frequency channels. What is the advantage of the windowed-cosine approach over a simple Discrete Fourier Transform (or is this a particular implementation of a DFT)? The description of the approach says that it aims to find "components that vary with different periods" - does this assume that the signals at each timescale will be periodic?

3) The computation of the timescale selectivity value (section 4.5) makes sense, but seems like it biases preferred timescales toward the middle of the timescale range, since it involves a weighted average across all timescales. There is also a particular (implicit) choice being made about how to convert from correlations r to the selectivity profile - after setting negative values to 0, this is a softmax with temperature = 1. But other choices could be made here: for example, using a lower temperature would concentrate the selectivity profile on the timescale with maximum r value (which would make the profile more unstable, but decrease the bias toward the middle of the timescale range). It would be helpful to see more justification of this step, and perhaps see how sensitive the results are to this choice of temperature.

4) The way that Figure 3 is introduced is difficult to understand: "a voxel with the same timescale selectivity for reading and listening could nevertheless have different selectivities to *individual* timescales between the two modalities" (lines 104-106). After reading through the Methods, I believe this is saying that Figs 1+2 are describing the timescale selectivity of each voxel with a single number (a weighted average of all timescales, weighted by the selectivity to each timescale) while Figure 3 is separately correlating the selectivity to each individual timescale. Is this the correct interpretation?

5) The fact that the same stories were used for reading and listening means that it should be possible to simply compute the timecourse correlation between the responses to the two modalities for each story, for each voxel. This would be helpful to see, since it would provide a model-free measure of response similarity across modalities, potentially identifying voxels in which the current model fails to find similar timescales across voxels but a different model could.

6) How is "Explainable Variance" computed in Supplementary Figure 6?

Minor comments:

7) "Timescale" here is defined in terms of a number of words, rather than time in seconds. These will obviously be highly correlated, but is it possible to instead apply the filters in the time (seconds) domain? Would there be any differences compared to the word-timescale hierarchy?

8) In section 4.4.4, I couldn't understand exactly which feature spaces and delays are going into the embeddings. There is a 9984-dimensional representation of each word, which is represented in 8 timescale filters, each delayed by 4 FIR delays? Are the sensory-level feature spaces (from 4.4.2) also being included in the regression? Are both sensory-level spaces used for both modalities, or does each modality get its own specific sensory-level features? Each feature space is given a separate regularization

parameter - what does that mean here (e.g. are the different FIR delays different feature spaces?)? The total number of feature spaces appears to be 10 based on section 4.4.5, so maybe the feature spaces are the 8 timescales and the 2 sensory-level features?

9) Due to the 120s high-pass filter applied during preprocessing, there should be some rough upper limit to the longest-timescale feature space - is there an approximate upper limit of the 256+ word filter?

10) Permutation testing was performed by block shuffling the target timecourse in blocks of 10 TRs. This is a reasonable approach (and has precedent in other work in this field) but the block length seems relatively short compared to the relevant timescales in this study (which go up to the paragraph+ scale). Would an alternative approach like circularly-shifting the target timecourse be more appropriate?

11) The authors do not provide a "Code Availability" statement/section. The technical approach used here is central to the paper, and it would be helpful (both for reproducibility as well as for other researchers hoping to apply this method to their own datasets) if the code could be publicly released, unless there is a compelling reason why this is not possible.

12) In Figure 1, at first I mistakenly thought that panels a and b were group data - the authors could make it more clear that these are results from S1.

13) There is a type-setting error on line 387: the 10^{10} exponent is not properly rendered.

Response to reviewer's comments: The Cortical Representation of Language Timescales is Shared between Reading and Listening

We thank all reviewers for their positive comments and constructive suggestions about our manuscript. We revised our manuscript according to the reviewers' comments and suggestions, and below we include point-by-point responses to each reviewer's questions.

In addition, we have removed the following sentences (originally on Lines 172-178 of the original Discussion). These sentences suggested that different experiments may be needed to observe differences between reading and listening. However, the listening and reading order was counterbalanced across subjects. Therefore, we think that an interleaved stimulus presentation could improve study design but do not think that it might lead to differences in timescale selectivity between the two modalities.

"A second limitation arises from the order of stimulus presentation. Each participant first read all eleven stories before listening, or vice versa. Although timescale selectivity was broadly consistent across participants regardless of which modality was presented first, future experiments with interleaved stimulus presentation modalities might identify small differences in timescale selectivity between the two modalities."

1 Reviewer #1 (Remarks to the Author):

In this manuscript, Chen et al., used voxelwise encoding models to investigate the topographic organization of language timescale representations across the cortical surface during both written and spoken language comprehension. The authors convincingly showed voxel selectivity to increasingly complex language processing timescales to form spatial gradients across prefrontal and temporal cortex, while also showing this topographic organization to be preserved across stimulus modality. The authors' analytic approach impressively corroborates and adds to earlier work on the subject while simultaneously correcting for previous limitations concerning timescale precision and ecological validity. However, we do have critiques concerning their use of BERT, interpretation of findings, and inconsistencies in figures.

Major Points:

Stimulus embedding:

Better justification is needed for the approach the authors took to constructing their stimulus embedding. In concatenating the activations of all 13 layers of BERT to get a p-dimensional ($p = 13 \times 768$) embedding per word, the authors cite three articles which supposedly point to this method as producing a "stimulus embedding [that] can explain a large proportion of the variance in brain responses to language stimuli." However, following closer inspection of the articles referenced as justification, they each seem to suggest that the use of embeddings derived from intermediate layers produces models which are most predictive of brain activity during spoken language comprehension, consistently outperforming models which use representations from the most peripheral layers. Indeed, it appears each study evaluates stimulus embeddings on a layer-by-layer basis (sometimes ignoring the input representation), ultimately reporting the brain predictivity of the model which used the best-performing layer. Nowhere, it seems, do they explicitly state the use of a stimulus embedding which concatenates the activations of all layers in tandem. Indeed, a surface reading suggests that incor-

porating the outermost layers into the stimulus embedding may confound the performance of their encoding models.

Still, this doesn't necessarily mean that including the activations from all of BERT's layers produces a poor stimulus embedding for their purposes; the authors just need to better justify why they took this approach. Because the first layer BERT receives is simply the sum of the token, segmentation, and position embeddings, the lower layers primarily represent linear word order, with linear/sequential information beginning to drop off at about layer 4. As such, as you advance up BERT's layers, BERT phases out positional information in favor of syntactically-sensitive hierarchical information, with syntactic information being most prominent in the representations of BERT's middle layers. Relative to syntactic information, semantic information is less localized to a given subset of layers but is best represented in higher layers. Indeed, the representations of deeper layers are required for BERT to handle difficult cases of subject-verb agreement that rely on long-range dependencies. The deepest layers of BERT are the most task specific. In the absence of fine-tuning, the activations of the final layer of the pre-trained BERT-based-uncased model the authors used are more specific to the masked language modeling (MLM) and next sentence prediction (NSP) tasks the model was trained on relative to earlier layers. All in all, BERT is said to have "rediscovered the classical NLP pipeline" as BERT, by and large, encodes basic syntactic features at the bottom-most layers (e.g., part-of-speech), high-level syntactic information in the middle layers (e.g., constituents), and complex semantic information at higher layers (e.g., semantic role labeling, coreference). Therefore, considering the authors are investigating the topographic representation of language structures of increasing complexity (e.g., "clause-level structures" → "paragraph-level structures"), perhaps concatenating the activations of all of BERT's layers can be justified as being an 'all-inclusive approach' that does not favor layer representations biased to a given timescale. Indeed, we suspect a reason why the three papers the authors cited found the intermediate layer representations of transformers to be best at predicting brain activity has to do with the fact that the middle layers of transformer-based language models tend to be the most transferable across a diverse range of NLP 'probing' tasks; i.e., the representations of intermediate layers tend to generalize the most across language timescales and therefore may be more predictive of voxel activity on average.

Nevertheless, work which seeks to identify what kinds of linguistic information are represented most robustly and transparently in each of (and across) BERT's layers is ongoing and does not suggest that knowledge of different language timescales is equally captured in the full model representation. Their use of linear filters to acquire timescale-specific features is intriguing and appears to have produced robust results, but it is not obvious to what extent the pre-filtered contextualized embeddings themselves differentially represent information of different language timescales. **Ultimately, we believe the approach the authors took to constructing their stimulus embedding may be reasonable but needs to be better justified**, including a discussion of possible confounds in their approach. Additionally, while we do not presume that their chief claim concerning the shared topographic organization of language timescale representations between written and spoken language comprehension will need modification, **exploring how this conclusion holds when experimenting with different stimulus embeddings (e.g., using just the intermediate BERT layers) seems to us to be a necessary control.**

We included embeddings from all layers of BERT in order to include as much stimulus information as possible in the stimulus features. By concatenating embeddings from all layers, we avoid biasing estimates of timescale selectivity towards the timescales that are preferentially included in a particular layer of BERT.

Nevertheless, we agree that the reviewer's suggestion of exploring how our conclusions hold across different layers of BERT is an important and informative analysis. Therefore, we performed additional analyses to estimate timescale selectivity from each layer of BERT separately. This resulted in 13 additional estimates of timescale selectivity (from the 12 hidden layers and one output layer of BERT).

In Figure R1 we compare estimates of timescale selectivity from each layer of BERT. We also compare to our original estimates, in which we used all layers of BERT together. Results are shown for one representative participant (S1; S2 showed qualitatively similar results but is omitted for space), for reading and listening separately. Figure R1A shows the number of voxels that are significantly predicted when using each BERT layer. The number of significantly predicted voxels in our original encoding models (i.e., with features derived from the concatenation of all layers) is

Fig. R1: **Comparison of embedding extraction methods, all layers of BERT vs only a single layer.** Timescale selectivity was estimated with embeddings from each layer of BERT. Results are shown for one representative participant. A. Bars denote number of significantly predicted voxels for each layer. Errorbars denote standard error. The dashed line denotes the number of significantly predicted voxels obtained from using all layers of BERT together. Including all layers of BERT generally produces better predictions of brain responses than only using a single layer. B. Timescale selectivity estimated with embeddings from each layer separately. For each significantly predicted voxel, timescale selectivity is shown according to the color scale at the bottom. Voxels that were not significantly predicted are shown in grey. (One-sided permutation test, $\alpha = 0.05$, FDR corrected.) Estimates of timescale selectivity are similar between the different embedding methods – spatial gradients from intermediate to long timescale selectivity are found along the superior to inferior axis of temporal cortex and along the posterior to anterior axis of prefrontal cortex (PFC), and precuneus is predominantly selective to long timescales. Estimates of timescale selectivity are robust to the choice of embedding extraction method. Embeddings from only a single layer of BERT often result in slightly worse prediction performance, but does not substantially affect estimates of timescale selectivity.

shown by the dashed horizontal line. The best layer varies across experimental conditions, and using all layers is as accurate as using the single best layer. Figure R1B shows the estimated timescale selectivity for each layer on the flattened cortical surface of the participant. Estimates of timescale selectivity are strikingly similar between different layers – spatial gradients from intermediate to long timescale selectivity are found along the superior to inferior axis of temporal cortex and along the posterior to anterior axis of prefrontal cortex (PFC), and precuneus is predominantly selective to long timescales. Although certain layers produce encoding models with worse prediction performance, estimates of timescale selectivity do not differ substantially between layers. This similarity suggests that our estimates of timescale selectivity are robust to the choice of embedding layer.

As suggested by the reviewer, we have revised the manuscript to clarify the motivation for our method for constructing stimulus embeddings. The revised Methods now includes the following description on Lines 374-375:

“Activation from units in all layers were included in order to include stimulus information at different levels of the language processing hierarchy.”

We have also added Figure R1 as Supplementary Figure S9.

RSVP reading: We understand that to make reading and listening more comparable, each word during the reading session was presented for a duration equal to the rate of that word’s presentation in the listening session. While matching the timing of word presentation across stimulus modalities is a good control, we worry that RSVP reading may be more similar to natural listening than natural reading, which would of course confound the claim that the cortical representation of language timescales is shared between reading and listening. Not only is RSVP reading not self-paced, but given each word is presented individually, participants are forced to read linearly and cannot look to neighboring words at any given time. We speculate that reading in this fashion may incline participants to rely on an internal monologue, verbalizing each word in their mind, more so than they would during natural reading. That is, **participants may be ‘listening’ to their own inner voice more than they normally would during natural reading and thus be activating auditory cortex-based mechanisms more than during natural reading. Conclusions about shared mechanisms across modalities may then be over-stated (including in the title) and need to be appropriately qualified.**

We agree that it is important to check whether the RSVP reading experiment activated auditory cortex. Therefore we tested whether low-level sensory features (i.e., spectrotemporal representations of stimulus audio recordings, and motion energy representations of stimulus visual presentations) predicted well in different low-level sensory cortical areas for reading and listening. As a control, we also compared the prediction performance of linguistic features (i.e., timescale-specific representations of the stimulus words).

Figure R2 compares the prediction performance of the low-level sensory features and of the linguistic features. Voxels are colored according to prediction performance: voxels shown in orange are well predicted by the low-level sensory features, voxels shown in blue are well predicted by the linguistic features, and voxels shown in white are well predicted by both feature spaces. Low-level sensory feature spaces predict well in different areas for the two stimulus modalities: low-level sensory features predict well in early visual cortex (EVC) during reading, and in early auditory cortex (AC) during listening. (For both reading and listening, linguistic feature spaces predict well in similar cortical areas.) These results suggest that the listening experiment activated early auditory cortex, whereas the reading experiment activated early visual cortex but not early auditory cortex. These results validate the RSVP reading paradigm and suggest that in the RSVP reading experiment, participants did not heavily rely on “listening” to an internal monologue.

We have revised the manuscript to include this comparison in the revised Figure 1D. (Please see our response to the next question for further details about this updated figure).

Within- vs. between-modality differences: The authors do a good job demonstrating, at both the individual and group level, that there exists a topographic organization of language processing timescales that is shared between reading and listening. While the paper’s focus is of course on cross-modal similarities, we believe that the authors do not sufficiently emphasize the differences in timescale representation between the stimulus modalities. **Several figures appear to show differences between modalities in short timescale representations in unimodal sensory cortices.**

Fig. R2: **Prediction performance of linguistic vs low-level sensory representations.** Prediction performance is shown on the flattened cortical surface of two representative participants, for reading and listening separately. Orange voxels were well-predicted by low-level sensory features. Blue voxels were well-predicted by linguistic features. White voxels were well-predicted by both sets of feature spaces. Low-level sensory features preferentially predict well in early visual cortex (EVC) during reading, and in early auditory cortex (AC) during listening. Linguistic features predict well in similar areas for reading and listening. These results suggest that the listening and reading experiments engaged different lower-level sensory cortices.

However, these differences are never explicitly discussed or measured in the results. Also, when the authors claim that timescale selectivity is significantly positively correlated between reading and listening, they only consider voxels that are language-selective to both modalities. As a consequence, in both the discussion and statistical analysis, modality-invariant language representations appear to be overemphasized, understating the differences in timescale selectivity between reading and listening and perhaps statistically overinflating the extent of similarity. Doing so may even serve to support the ecological validity of their RSVP reading paradigm by showing how the reading paradigm does indeed engage different areas from the listening paradigm.

We agree that the differences in other cortical areas should be more clearly acknowledged in the text. We have added a new panel Figure 1D to highlight differences in language representations that occur in unimodal sensory cortices. The revised Results now states on Lines 93-105:

"In contrast to representations of language timescales, low-level sensory features are represented in modality-specific cortical areas. Figure 1D shows the prediction performance of linguistic features (i.e., timescale-specific feature spaces), and the prediction performance of low-level sensory features (i.e., spectrotemporal representations of stimulus audio recordings, and motion energy representations of stimulus visual presentations). Voxels are colored according to the prediction performance: voxels shown in blue are well predicted by the linguistic features, voxels shown in orange are well predicted by the low-level sensory features, and voxels shown in white are well predicted by both feature spaces. For both reading and listening, timescale-specific feature spaces predict well broadly across temporal, parietal, and prefrontal cortices. In contrast, low-level stimulus features predict well in early visual cortex (EVC) during reading only, and in auditory cortex (AC) during listening only. These results indicate that during language comprehension, linguistic stimulus features are processed in similar cortical areas between modalities, whereas low-level sensory processing occurs in modality-specific cortical areas."

Moreover, we have revised the introduction of our manuscript to clarify our motivation in comparing timescale selectivity across voxels that are language-selective in both modalities. The revised Introduction now states on Lines 34-45:

"At low levels of the hierarchy, brain representations are known to differ between the two stimulus modalities. For example, visual letterforms in written language are represented in the early visual cortex, whereas articulatory features in spoken language are represented in the early auditory cortex (Heilbron et al., 2020; de Heer et al., 2017). In contrast, many parts of temporal, parietal, and prefrontal cortices process both written and spoken language (e.g., Booth et al., 2002; Buchweitz et al., 2009; Liuzzi et al., 2017; Regev et al., 2013; Deniz et al., 2019; Nakai et al., 2021). It could therefore be the case that in these areas representations of higher-level language components are organized in the same way for both written and spoken language comprehension. On the other hand, these areas could contain overlapping but independent representations for the two modalities. One way to differentiate between these two possibilities would be to directly compare the cortical organization of brain representations across high-level language components between reading and listening."

Sequence input context length: It may be worthwhile for the authors to provide justification as to why they provided one sentence at a time as input, when BERT-base models handle a maximum input length of 512 tokens. Considering a goal is to extract timescale features related to long-range dependencies between distant tokens, it isn't immediately obvious to us as to why the authors wouldn't extend their context length window to be longer, or experiment with different input sequence lengths, instead of settling on word embeddings which are only locally contextualized. In our quick reading of the Jain and Huth (2018) paper that's referenced in the manuscript, an appropriate justification may be that voxelwise encoding model performance increases monotonically as a function of context length, hitting a point of diminishing returns around the length of an average sentence. However, the J&H paper only examines this relationship up to a context length of 20 and uses a LSTM language model. Nevertheless, the Toneva and Wehbe (2019) paper the authors cited uses BERT and found that encoding model performance (voxel prediction accuracy) improves monotonically up to about

15 words (depending on the layer) and drops off after as context length increases (at least up to 40 words). Therefore, feeding BERT one sentence at a time may be reasonable, especially when considering that the complexity of BERT's attention mechanism is quadratic with respect to context length; however, the authors need to make their reasoning explicit.

Be that as it may, the J&H (2018) study found that language timescale selectivity can be probed, albeit in a rather confounded and uninterpretable way, by varying context length. They found that low-level language areas (e.g., auditory cortex) are selective to contextual LSTM representations built from short sequence inputs, whereas higher-language areas prefer LSTM embeddings built from inputs of longer lengths which incorporate more context. It is unclear to us whether the authors' method of extracting timescale-specific features is confounded by using a rather short context length, especially since sentences are themselves relatively self-contained units of meaning whereas J&H (2018) appear to treat their input sequences as an arbitrary span of contiguous text; however, we believe it is worth investigating.

Ultimately, we believe it necessary that the authors evaluate how their conclusions hold when systematically varying the context length (i.e., number of tokens) passed to BERT-base for two reasons: (1) timescale representations of language in the brain have been shown to be sensitive to the amount of context given to language models, and (2) it is unclear whether the longer-range timescale features fail to adequately represent long-range information as they are extracted from stimulus embeddings that are only contextualized in the short to intermediate range. We anticipate that varying context length will not result in the primary claims of the manuscript needing to be significantly modified, but firmly believe that this still needs to be explicitly tested given how the authors' methods differ from previous studies.

We appreciate the reviewer's careful consideration and agree that exploring different context lengths would further validate our method. To explore whether our results are consistent across different context lengths, we compared two different methods of extracting embeddings from BERT. Both methods involved a rolling input context, in which the embedding of each word w_i was extracted based on the input context $w_{i-10} \dots w_i$. In one method, we used a short rolling input context length of $i = 10$ words. In the other method, we used a long rolling input context length of $i = 100$ words.

Figure R3 compares the two embedding extraction methods. Figure R3A compares the prediction performance of the two methods. Prediction performance is slightly higher with context length 10 than with context length 100 – using a longer context length of 100 words does not result in better model predictions. Figure R3B shows voxelwise estimates of timescale selectivity are shown for two representative participants. Estimates of timescale selectivity are slightly longer in some areas when a 100-word context is used than when a 10-word context is used (e.g., voxels in inferior temporal cortex are darker red in the 100-word context than with the 10-word context). However, overall the organization of timescale selectivity is qualitatively similar across these two embedding extraction methods. These results indicate that estimates of timescale selectivity are robust to the context length passed to BERT-base.

To demonstrate the robustness of estimated timescale selectivity to different context lengths, we have added Figure R3 to the manuscript as Supplementary Figure S8.

Minor Points:

Mismatch in timescale selectivity patterns between the flattened cortical surface and inflated cortex: In Figure 2B, the activity patterns shown on the flattened versus inflated cortical representations do not appear to match. The inflated representations have a lot of mid-level selective voxels (green/blue) in lateral occipital cortex whereas the flattened ones do not. This appears to be because the authors portrayed vertices that were significantly predicted for both modalities (in at least 1/3 of the participants) on the flattened cortical surface, but instead included vertices specifically predicted within-modality on the inflated cortex. This figure and the others should be checked for accuracy.

We thank the reviewer for pointing out this inaccuracy. This inaccuracy was due to a coding error when saving the inflated brain views. We have fixed Figure 2B and have checked other figures for accuracy.

We have replaced Figure 2B. The old and new versions are shown in Figure R4.

Map timescale selectivity patterns against known networks: Given that the data have been projected to a cortical surface, the authors could easily take the extra step to plot the voxelwise results

Fig. R3: Comparison of embedding extraction methods, input context length 10 words vs input context length 100 words. A. Encoding model prediction performance (r) using a rolling context of 10 words (x-axis) vs a rolling context of 100 words (y-axis). Each point represents one voxel. Axis labels indicate the number of voxels for which the respective embedding extraction method produces better performance (Some voxels are predicted similarly well with both embedding extraction methods; therefore, percentages may not sum to 100.) An input context length of 10 words produces more accurate predictions of brain responses than only an input context length of 100 words. B. Timescale selectivity is shown for two representative participants, for reading and listening separately. For each significantly predicted voxel, timescale selectivity is shown according to the color scale at the bottom. Voxels that were not significantly predicted are shown in grey. (One-sided permutation test, $\alpha = 0.05$, FDR corrected.) For both embedding extraction methods, temporal cortex contains a spatial gradient from intermediate to long timescale selectivity along the superior to inferior axis, prefrontal cortex (PFC) contains a spatial gradient from intermediate to long timescale selectivity along the posterior to anterior axis, and precuneus is predominantly selective to long timescales. These results suggest using a shorter input context length produces more accurate models of brain responses, and that estimates of timescale selectivity are robust to the choice of input context length.

against known brain networks. In particular, showing whether the long-timescale voxels map onto the default mode and cognitive control networks would help contextualize the findings in terms of other studies that have looked at, for instance, how the default mode network is involved in theory-of-mind operations. In addition, some of the transitions between short and long-timescale representations look more categorical than gradual and may correspond to known borders separating auditory, lan-

Fig. R4: Old vs new version of Figure 2B

guage, and default-mode networks. Similarly, plotting the results against language atlases (i.e., the ones available from the Fedorenko lab) will help put these results in the context of the known language network architecture. Doing this will not only improve readability but also reliability, as other researchers' results can be compared in the same anatomical framework.

To compare our estimates of timescale selectivity to previously proposed cortical networks, we used the Yeo2011 cortical parcellation (Yeo et al., 2011). We used the 17-network parcellation, and then used pre-defined network labels to group these parcels into proposed networks. We investigated timescale selectivity for three of these proposed networks: Temporal-Parietal (Temp-Par), Cognitive Control (Control), and Default Mode (DMN) networks. We focus on these three networks because they have high overlap with the set of vertices that are well-predicted by the timescale-specific feature spaces. Collectively, these three networks include 90% of vertices that are significantly predicted in at least three participants for both modalities.

Figure R5A shows group-level estimates of timescale selectivity within each of the three networks. Each network displays selectivity to a range of different timescales, and gradients of timescale selectivity are evident within each network. For instance, the tempo-parietal network shows a gradient from short to long timescale selectivity in superior to inferior temporal cortex, and a mix of timescale selectivity is shown in prefrontal areas of the control network. The DMN shows a gradient from short to long timescale selectivity from posterior to anterior prefrontal cortex. Figure R5B shows the distribution of timescale selectivity for each network. Variation in timescale selectivity within each network is consistent with prior work suggesting that language processing occurs along a continuous gradient, rather than in distinct, functionally specialized brain networks (Blank and Fedorenko, 2020).

Furthermore, the DMN contains longer timescale selectivity than the other networks, for both reading and listening, at both the group level and in each individual participant. The statistical significance of this difference was tested with an independent t-test for samples with unequal variance. This test was conducted separately for reading and listening, separately for the group level and for each individual participant. This difference was statistically significant for both reading and listening, at the group level and for eight of the nine individual participants ($p = 0.13$ for DMN vs TempPar S5 reading; $p < 0.01$ at the group level and for all other participants, modalities, and network pairs). The preference towards longer timescale selectivity within the DMN is consistent with reports that long-timescale narrative-length information may be processed in this network (Simony et al., 2016).

We have revised the manuscript to include this comparison to known anatomical networks. We

Fig. R5: Group-level timescale selectivity across previously proposed cortical networks. The Yeo2011 cortical parcellation was used to determine 17 cortical parcels in fsaverage space (Yeo et al., 2011), and then from this parcellation pre-defined network labels were used to identify three proposed networks (temporo-parietal network (TempPar), cognitive control network (Control), and default mode network (DMN)). **A.** Group-level timescale selectivity is shown on the flattened cortical surface of the fsAverage template brain, for the three networks separately, and for reading and listening separately. Timescale selectivity is shown according to the color scale at the bottom. Voxels that were not significantly predicted for both modalities in at least three participants are shown in grey. (One-sided permutation test, $\alpha = 0.05$, FDR corrected.) The temporo-parietal network (TempPar) contains a gradient from short to long timescale selectivity in superior to inferior temporal cortex. Prefrontal areas of the control network (Control) contain a mix of timescale selectivity. The default mode network (DMN) contains a gradient from short to long timescale selectivity from posterior to anterior prefrontal cortex. **B.** The distribution of group-level timescale selectivity for each network is shown for reading and listening separately. Histograms include vertices that were significantly predicted for both modalities in at least three participants. Each network is selective to a range of different timescales. The DMN contains longer timescale selectivity than the other networks. This difference was statistically significant for both reading and listening, at the group level and for eight of the nine individual participants ($p = 0.13$ for DMN vs TempPar S5 reading; $p < 0.01$ at the group level and for all other participants, modalities, and network pairs).

have included Figure R5 as Supplementary Figure S6 in the manuscript.

2 Reviewer #2 (Remarks to the Author):

This work investigates the timescales of language representations during listening and reading of the same naturalistic story. The authors use voxelwise encoding models to predict fMRI recordings as a function of timescale representations that are derived from a spectral decomposition of contextualized word embeddings from a large language model (BERT). The main claim in the work is that the timescales of language representations during reading and listening are the same, based on results showing significant correlations between the timescale selectivity during listening vs reading for a large number of voxels that are typically associated with language comprehension.

In general, I believe the approach proposed by the authors has promise. However, I am not yet convinced of their claim because of concerns relating to their methodology and analysis choices.

Methodology questions:

1. **Are the features at different timescales correlated with one another?** Such correlations would impede disentangling these features's contributions to the brain prediction. **How does the split-prediction performance method that the authors use deal with such correlations?**

Features at different timescales had low correlation with each other – for each pair of feature spaces, the mean pairwise correlation coefficient between dimensions of the feature spaces was less than 0.1. Our split-prediction performance measures are derived from banded ridge regression. In banded ridge regression, the hyperparameter selection optimization acts as a feature-selection mechanism that emphasizes features that predict well, and de-emphasizes features that are redundant. This feature-selection mechanism has been validated with simulated data and with recorded fMRI data (Dupré la Tour et al., 2022). In particular, Figure 3 of Dupré la Tour et al. (2022) uses simulated examples to show that in situations with correlated feature spaces, banded ridge regression properly recovers the feature spaces that explain variance in brain responses. In these examples, simulated feature spaces were used to generate simulated signals. Some feature spaces were correlated with other feature spaces, and some feature spaces were not truly predictive of the generated signals. Whereas standard ridge regression selected non-predictive feature spaces and incorrectly ignored correlated feature spaces, banded ridge regression correctly ignored non-predictive feature spaces and recovered the correct variance decomposition between redundant feature spaces. Thus, using banded ridge regression helps to account for stimulus feature correlations.

We have revised the Methods to state on Lines 435-437:

Note that the resulting feature spaces had low correlation with each other – for each pair of feature spaces, the mean pairwise correlation coefficient between dimensions of the feature spaces was less than 0.1.

We have also revised the Methods to state on Lines 480-481:

"Note that hyperparameter selection in banded ridge regression acts as a feature-selection mechanism that helps account for stimulus feature correlations (Dupré la Tour et al., 2022)."

2. **Doesn't the current formulation of timescale selectivity equate cases where there is very peaked selectivity for one frequency band, and cases where there is more uniform selectivity across frequency bands?** I would think that we want to have a way to distinguish those cases.

"Average timescale selectivity" does indeed equate these two cases. However, these two cases can be distinguished by the "timescale selectivity profile." For each voxel, the timescale selectivity profile reflects the selectivity to each of the eight timescales.

We have revised the manuscript to emphasize that we compare the timescale selectivity profile in order to disambiguate between these two cases.

The revised Results now states on Lines 135-138:

"However, average timescale selectivity could equate two types of voxels: voxels with a very peaked selectivity for a single frequency band, and voxels with uniform selectivity to many frequency bands. To investigate this possibility we used the timescale selectivity profile, which reflects selectivity to each timescale separately."

3. **It's not clear how the contextualized word representations were extracted from BERT.** Was

a masked token used? Was the representation at each word in the input sentence extracted, or was it extracted only at the last token? What was done with words that have multiple tokens? BERT is a bidirectional model, so if the representations of all words were extracted instead of just at the last word, then **information about future words is included in many of the word representations. Can the authors comment on this choice and how they think it affects their results?**

To extract contextualized word embeddings, the stimulus text was input to BERT sentence-by-sentence. The embedding of each word was extracted from this bidirectional context. For words with multiple tokens, we used the mean over the token embeddings. We chose sentence-length inputs because sentences provide more natural splits than a fixed rolling context length. Mask tokens were not used.

To address the reviewer's concern we compared our original method that used sentence-level input contexts to extract embeddings, to an alternate method of using a fixed rolling input context length of 10 words. In the alternate rolling input context method, for each word w_i of the stimulus narratives, only the preceding words $w_{i-10} \dots w_i$ were input to BERT, and then the embedding of word w_i was extracted based on this rolling context.

Figure R6 compares these two embedding extraction methods. Results are shown for two representative participants (S1 and S2) and for reading and listening separately. Figure R6A compares prediction performance between the sentence-level context (x-axis), and the rolling context (y-axis). Each point reflects the prediction performance of one voxel. Prediction performance is consistently higher with sentence-level context than with the rolling context of 10 words. Therefore, we retain the use of sentence-level inputs in our main manuscript. Figure R6B shows estimates of timescale selectivity for each embedding extraction method on the flattened cortical surface of each participant. The organization of timescale selectivity is qualitatively similar across embedding extraction methods – estimates of timescale selectivity derived from both methods show spatial gradients from intermediate to long timescale selectivity along the superior to inferior axis of temporal cortex and along the posterior to anterior axis of prefrontal cortex (PFC), and long timescale selectivity in precuneus. This similarity suggests that estimates of timescale selectivity are robust to the embedding extraction method – including information about future results during embedding extraction does not substantially affect estimates of timescale selectivity.

We have added Figure R6 as Supplementary Figure S7.

Additional analyses that would strengthen the claims:

4. **How do the results align with those from Lerner et al. 2011 that suggest that several language regions are sensitive to short and long timescales? In general, I would like to better understand the estimated selectivity across timescales – is it mostly peaked at one timescale, or is uniform across several close timescales, and is there some systematic shape in the distribution of selectivity across regions?** From the Lerner et al. 2011 results, I would expect that the STS would have a more uniform distribution of selectivity, whereas the PC would have a peaked distribution at the long timescales.

To explore this question, we measured the uniformity of timescale selectivity across timescales. We computed the entropy of the timescale selectivity profile for each voxel. The entropy was defined as $\sum_{i=1}^8 (\tilde{r}_i^t \log_2(\tilde{r}_i^t))$, where \tilde{r}_i^t refers to the i 'th index of the timescale selectivity profile. Voxels with more uniform timescale selectivity have a flatter \tilde{r}^t distribution; thus, higher entropy corresponds to more uniform timescale selectivity, whereas lower entropy corresponds to more peaked timescale selectivity.

Figure R7 shows the entropy of voxel timescale selectivity profiles for two representative participants, for reading and listening separately. Voxels have more uniform (i.e., higher entropy) timescale selectivity profiles in STG and posterior areas of prefrontal cortex, and are more peaked (i.e., lower entropy) in lateral and medial parietal cortex. These results are consistent with the reviewer's expectation and with Lerner et al. (2011), which found that areas near STS and posterior PFC displayed a wider range of temporal receptive windows than other brain areas, whereas areas such as precuneus displayed a smaller range of temporal receptive windows.

We have added a figure describing the uniformity of timescale selectivity as Supplementary Figure S5.

5. Re Fig 3 results: it's strange to compute correlations across all voxels for reading vs listening

Fig. R6: Comparison of embedding extraction methods, sentence input context vs fixed rolling input context. A. Encoding model prediction performance (r) obtained from a sentence-split input context (x-axis), and from a rolling context of 10 words (y-axis). Each point represents one voxel. Axis labels indicate the number of voxels for which the respective embedding extraction method produces better performance (Some voxels are predicted similarly well with both embedding extraction methods; therefore percentages may not sum to 100.) A sentence-split produces more accurate predictions of brain responses than a rolling input context length of 10 words. B. Timescale selectivity is shown for two representative participants and for reading and listening separately. Timescale selectivity is shown according to the color scale at the bottom. Voxels that were not significantly predicted are shown in grey. (One-sided permutation test, $\alpha = 0.05$, FDR corrected.) For both embedding extraction methods, temporal cortex contains a spatial gradient from intermediate to long timescale selectivity along the superior to inferior axis, prefrontal cortex (PFC) contains a spatial gradient from intermediate to long timescale selectivity along the posterior to anterior axis, and precuneus is predominantly selective to long timescales. While Panel A shows that a sentence-split input context length produces more accurate models of brain responses, estimates of timescale selectivity are robust to the input context method.

to indicate selectivity because the authors already showed in Fig 2 that there is a large difference across voxels! **I believe a much more straightforward analysis would be to correlate the vectors of timescale selectivity for listening vs reading, for each voxel.** That would yield a voxel-wise estimate of how similar the modality-specific selectivity profiles are.

To obtain a voxel-wise estimate of the similarity between timescale selectivity in each modality, we computed the correlation coefficient of the timescale selectivity profile between reading and

Fig. R7: **Uniformity of timescale selectivity.** The entropy of the timescale selectivity profile is shown on the flattened cortical surface of two representative participants, for reading and listening separately. The entropy of each voxel is shown according to the color scale at the bottom. Brighter voxels have higher-entropy (i.e., more uniform) timescale selectivity profiles. Darker voxels have lower-entropy (i.e., more peaked) timescale selectivity profiles. Voxels that were not significantly predicted are shown in grey. (One-sided permutation test, $\alpha = 0.05$, FDR corrected.) Voxel timescale selectivity profiles have higher entropy (i.e., are more uniform) in superior temporal gyrus (STG) and posterior areas of prefrontal cortex (PFC), and have lower entropy (i.e., are more peaked) in lateral and medial parietal cortex. These results are consistent with the reviewer’s expectation and with Lerner et al. (2011), which found that areas near STS and posterior PFC displayed a wider range of temporal receptive windows than other brain areas, whereas areas such as precuneus displayed a smaller range of temporal receptive windows.

listening. Figure R8 shows this correlation for each participant. The timescale selectivity profile is highly correlated across voxels in areas selective to both modalities.

We have included Figure R8 as Figure 3 in the revised manuscript and revised our manuscript to include this analysis. In the revised Results, Lines 141-145 now states:

“To directly compare timescale selectivity profile between modalities, for each voxel we computed the Pearson correlation coefficient between the timescale selectivity profile in reading and in listening. Figure 3 shows this correlation on the flattened cortical surface of each participant. The timescale selectivity profile is highly correlated between reading and listening, across voxels that are language-selective in both modalities.”

6. Do the authors believe the representations they construct capture all relevant language timescales for both modalities? For example, it doesn’t seem that many voxels prefer short timescales currently, but is it possible that BERT does not capture timescales that are short enough because it is a text-only model? **Would the selectivity results of listening vs reading be different given representations from an audio model, such as wav2vec, rather than a text-only model?**

We do not believe that our timescale-specific features capture all relevant language information. For instance, audio models such as wav2vec incorporate additional features that vary faster than single words, such as sound spectrograms and articulatory features. Prior studies showed that audio models can indeed estimate representations in the auditory cortex, but these models produce much poorer brain prediction in higher-level cortical areas (e.g., Millet et al. (2022) Figure 2). However, estimates of timescale selectivity would be much poorer for higher-level brain areas. Here we were interested in studying timescale selectivity for linguistic features, rather than for sensory-level auditory or visual features; therefore, we used text-only models that have better representations of language at the level of words and above.

Fig. R8: **Voxelwise similarity of timescale selectivity.** The Pearson correlation coefficient of the timescale selectivity profile between reading and listening is shown on the cortical surfaces of each participant. The correlation coefficient is shown according to the color scale at the bottom. Red voxels have positively correlated timescale selectivity profiles between reading and listening. Blue voxels have negatively correlated timescale selectivity profiles between reading and listening. Voxels that were not significantly predicted in both modalities are shown in grey. (One-sided permutation test, $\alpha = 0.05$, FDR corrected.) The timescale selectivity profile is highly correlated across voxels in areas selective to both modalities.

We indeed think the reviewer's question is very relevant, and revised the manuscript to emphasize that our features do not capture all relevant language information. Lines 248-251 of the revised Discussion states:

"A second limitation arises from the current state of language model embeddings. Although embeddings from language models explain a large proportion of variance in brain responses, these embeddings do not capture all stimulus features (e.g., features that change within single words)."

We have also revised the manuscript to discuss the possibility of applying our method to low-level auditory and visual features. Lines 234-236 of the revised Discussion states:

"In the future, our method could be used with pretrained audio or visual models (e.g., wav2vec 2.0 (Baevski et al., 2020) or TrOCR (Li et al., 2023)) to estimate selectivity to different timescales of low-level auditory and visual features."

Better discussion of related work:

7. The authors claim that "prior work has not studied the organization of timescale representations during written language comprehension". **Some related work that should be discussed here is that of Toneva and Wehbe 2019**, which is already referenced by the authors, but should also be further discussed in light of this claim. This previous work shows that, during written language comprehension, brain regions that are thought to be sensitive to long timescales (e.g. AG, PC, dmPFC; Lerner et al. 2011) are better predicted by contextualized word embeddings with longer contexts than by non-contextualized word embeddings. In contrast, regions that are thought to be sensitive to both short and long timescales (e.g. STS, IFG) are predicted equally well by non-contextualized as well as contextualized embeddings.

We agree that the results in Toneva and Wehbe 2019 are highly relevant and deserve more in-depth discussion. We have revised our manuscript to better contextualize our results with respect to those in Toneva and Wehbe 2019.

The revised Discussion now includes the following discussion on Lines 181-188:

"Prior studies showed differences in non-contextualized and contextualized brain representations for written and spoken language comprehension separately (Toneva and Wehbe, 2019; Jain and Huth, 2018). These results showed that areas within medial parietal cortex, prefrontal cortex, and inferior temporal cortex preferentially represent contextualized information; whereas other areas within superior temporal cortex and the temporoparietal junction do not show a preference for contextualized information. Our results build upon these previous findings by examining representations across a finer granularity of timescales, and by directly comparing representations between reading and listening within individual participants."

In addition, we have revised our manuscript to clarify the state of prior work. The original manuscript stated on Lines 45-49:

"Although prior studies have estimated the topographic organization of timescale representations in the brain during spoken language comprehension (Lerner et al., 2011; Baldassano et al., 2017; Blank and Fedorenko, 2020; Jain and Huth, 2018; Jain et al., 2020), prior work has not studied the organization of timescale representations during written language comprehension."

The revised Introduction states on Lines 48-54:

"Other studies focused on relatively few components (e.g., low-level sensory features, word-level semantics, and phonemic features), and therefore did not provide a detailed differentiation between representations of different levels of the language hierarchy (Deniz et al., 2019; Nakai et al., 2021). Studies that did differentiate between different levels focused on one modality of language (e.g., Toneva and Wehbe, 2019; Lerner et al., 2011; Jain and Huth, 2018; Jain et al., 2020). Prior studies are therefore insufficient to determine whether brain representations of the language hierarchy are organized similarly between reading and listening."

8. Additionally, **Oota et al. 2022 <https://aclanthology.org/2022.naacl-main.235.pdf> investigate the voxelwise encoding performance during both listening and reading, and find differences in which task-optimized models predict each modality well.** One could argue that the different tasks that these models are optimized for may relate to different timescales of language. Oota and colleagues use two datasets corresponding to different stimuli so it's harder to make direct claims about differences between reading and listening, but **the authors should at least discuss this previous work since their results can be seen to disagree with these previous findings.**

We hypothesize that the differences between our results and those in Oota et al. (2022) are indeed because Oota et al. (2022) used reading and listening data corresponding to different stimuli. The reading data used sentence-level stimuli, whereas the listening data used narrative-level stimuli. Prior work has shown that the amount of stimulus context heavily influences brain representations (Deniz et al., 2023). This difference in stimulus context could explain the differences reported in Oota et al. (2022) between reading and listening responses. Indeed, this difference could explain why listening data were better predicted by models trained on higher-level tasks (e.g., summarization and paraphrase detection).

We have revised our manuscript to discuss distinctions between our work and Oota et al. (2022). The revised Discussion states on Lines 204-212:

"Another study suggested that different types of language representations explain brain responses to written and spoken language (Oota et al., 2022). However, in Oota et al. (2022) the reading stimuli contained higher-level information than the listening stimuli: the stimuli used for reading experiments consisted of isolated sentences, whereas the stimuli used for listening experiments consisted of full narratives. This discrepancy perhaps explains why in Oota et al. (2022), language models trained on higher-level tasks (e.g., summarization, paraphrase detection) were better able to predict listening than reading data. Indeed, prior work has shown that longer stimulus context results in stronger language representations in the brain (Deniz et al., 2023). This discrepancy supports the importance of using narrative-length, naturalistic stimuli to elicit brain representations of high-level linguistic features."

Better explaining or toning down some statements:

9. **L102: "These group-level results indicate that across participants, representations of language timescales are organized in the same way between reading and listening." This statement is too strong, given both the methods and the results.** What the results show is that the timescales *that are captured in the constructed representations* *align significantly* between the two modalities. There can be other timescales that are not captured by the current representations that do not align between the two modalities.

We agree that the original statement that representations are organized "in the same way" was too strongly worded. Our results show that the timescales align significantly, and that the prediction accuracy across timescales was similar between reading and listening. However, our features do not capture all relevant stimulus information.

We have revised the manuscript to rephrase L102. The original Results stated on Lines 102-103:

"These group-level results indicate that across participants, representations of language timescales are organized in the same way between reading and listening."

The revised Results states on Lines 131-133:

"Overall, these group-level results show that across participants, the organization of representations of language timescales is consistent between reading and listening."

Moreover, the revised manuscript includes a discussion of the limitations of stimulus features captured by our stimulus embeddings. The revised Discussion states on Lines 249-253:

"Although embeddings from language models explain a large proportion of variance in brain responses, these embeddings do not capture all stimulus features (e.g., features that change within single words). In the future, our method can be used with other language models to obtain more accurate estimates of timescale selectivity than estimates from current language models."

10. L39: However, it is unclear whether within these cortical areas written and spoken language elicit the same or different topographic organizations of language timescale representations. The answer to this question would suggest whether the same pathways are used to integrate written and spoken language. I don't quite follow the logical implications here. **Why would similar topographic organizations of language timescale suggest that the same pathways are being used to integrate written and spoken language?** Perhaps it is slightly more granular information than just showing that the same regions are involved in language processing from both modalities, but I fail to see how it's conclusive evidence that the same pathways are involved.

We apologize for that our original statement was unclear. Comparing the topographic organization of language timescales is not conclusive evidence that the same pathways are involved, but rather a step towards understanding whether written and spoken language processing generate similar representations of different stages of the language processing hierarchy.

We have revised the manuscript to clarify our claim and motivation. The original Introduction stated on Lines 39-41:

"However, it is unclear whether within these cortical areas written and spoken language elicit the same or different topographic organizations of language timescale representations. The answer to this question would suggest whether the same pathways are used to integrate written and spoken language."

The revised Introduction states on Lines 40-45:

"It could therefore be the case that in these areas representations of higher-level language components are organized in the same way for both written and spoken language comprehension. On the other hand, these areas could contain overlapping but independent representations for the two modalities. One way to differentiate between these two possibilities would be to directly compare the cortical organization of brain representations across high-level language components between reading and listening."

Minor:

The writing is for the most part clear and concise. However, **there needs to be some description of what the timescale selectivity representation is in the main text.** Currently, it is very difficult to understand the results without actually reading the methods pretty thoroughly.

We have revised our manuscript to more clearly describe timescale selectivity.

The revised Introduction now states on Lines 56-63:

"Intuitively, levels of processing hierarchy can be considered in terms of numbers of words. For example, low-level sensory components such as visual letterforms in written language and articulatory features in spoken language vary within the course of single words; sentence-level syntax varies over the course of tens of words; paragraph-level semantics varies over the course of hundreds of words. Therefore we operationalize the notion of levels of the language hierarchy as language timescales, where a language timescale is defined as the set of spectral components of a language stimulus that vary over a certain number of words. For brevity we refer to "language timescales" simply as timescales."

[...]

Voxelwise encoding models were used to estimate the average timescale to which each voxel is selective, which we refer to as the "average timescale selectivity."

L 192: the reference should appear in parentheses

We thank the reviewer for pointing this out. We have revised the manuscript accordingly.

3 Reviewer #3 (Remarks to the Author):

In this study the authors make use of an fMRI dataset in which subjects read and listened to narratives. They propose a novel method for building a mapping from words to neural responses (using a pre-trained Natural Language Processing transformer model) and then decomposing this mapping into different timescales. This allows them to identify the timescale of stimulus information that each voxel is sensitive to, without requiring experimental manipulations like scrambling stimulus content. They find that there is a clear hierarchy of response timescales throughout the brain, and that this timescale is largely similar between reading and listening.

This is a novel and elegant approach to identifying timescales of responses to language, which connects to but also extends previous work on naturalistic response timescales. The figures are simple but compelling, with patterns even in individual subjects. The finding that this timescale hierarchy is consistent across subjects *and* across modalities is very impressive, and provides an "atlas" that could be used in many future studies. This work should be of great interest to multiple subfields of cognitive neuroscience, including language perception, naturalistic dynamics, and neuroimaging methods.

My comments are primarily about the details of the novel analysis being proposed, which could use some additional clarification and justification.

Major comments:

1) The specific way that the word embeddings are being obtained involves a few analysis choices which could be better justified. Specifically:

a) **What is the motivation for using BERT specifically, rather than other language models?** Models trained on other tasks could potentially provide better (or different) feature embeddings. For example, cosine distances between the top-level embeddings from BERT have been shown to perform poorly on semantic similarity tasks (Reimers & Gurevych, 2019, arXiv:1908.10084), though that concern may not apply here, since regression weights are being learned on top of the embeddings.

We used BERT specifically, because in settings where regression weights are learned on top of the embeddings, embeddings from BERT have been shown to accurately predict brain responses (Toneva and Wehbe, 2019; Caucheteux and King, 2020; Schrimpf et al., 2021). We agree that embeddings from other models have better performance when directly used for downstream tasks such as semantic similarity, but that this concern does not apply here because we learn regression weights on top of the embeddings. In the future, our method can be used with more specialized language models to test whether task-specific training enhances estimates of certain timescales. We release our feature extraction code to enable this direction of future work.

We have revised the Discussion to highlight the possibility of applying our method to other models. Lines 249-253 of the revised manuscript states:

"Although embeddings from language models explain a large proportion of variance in brain responses, these embeddings do not capture all stimulus features (e.g., features that change within single words). In the future, our method can be used with other language models to obtain more accurate estimates of timescale selectivity than estimates from current language models."

b) **The authors compute word embeddings by putting one sentence at a time into BERT. Does this bias the features toward a sentence-level timescale?** E.g. does it limit the extent to which paragraph-level dependencies could appear in the feature space?

We appreciate the reviewer's careful consideration. To further validate our method, we explore whether different context lengths bias timescale selectivity towards certain timescales. We compared two different methods to extract embeddings from BERT: a short rolling input context length of 10 words (roughly corresponding to the length of a sentence), and a long rolling input context length of 100 words (roughly corresponding to the length of a paragraph). To extract a word embedding with a rolling input context of i words, each word w_i of the stimulus narratives was input to BERT along with words $w_{i-10}...w_i$. Then the embedding of word w_i was extracted based on this rolling context.

We also performed this analysis in response to a similar comment raised by Reviewer 1. Hence, we refer to Figure R3, which compares the two embedding extraction methods. Figure R3

compares the two embedding extraction methods. Figure R3A compares the prediction performance of the two embedding extraction methods. Prediction performance is slightly higher with context length 10 than with context length 100; using a longer context length of 100 words does not result in better model predictions. Voxelwise estimates of timescale selectivity are shown for two representative participants in Figure R3B shows voxelwise estimates of timescale selectivity are shown for two representative participants. Estimates of timescale selectivity are slightly longer in some areas when a 100-word context is used than when a 10-word context is used (for instance, voxels in inferior temporal cortex are darker red in the 100-word context than with the 10-word context). However, overall the organization of timescale selectivity is qualitatively similar across these two embedding extraction methods.

Moreover, using a rolling input context length of 10 words instead of a sentence-split context also produces similar estimates of timescale selectivity (please see Figure R6 for details). These similarities suggests that our input context does not meaningfully bias estimates of timescale selectivity. Nevertheless, in the future, with the development of language models that better represent long-term dependencies, our method can be used with those language models to further explore this question.

To demonstrate the consistency of estimates of timescale selectivity between different context lengths, we have added these comparisons in Supplementary Figure S8.

c) One unusual property of computing word embeddings in a bidirectional language model is that the feature representation of a word can depend on words that come after it in the sentence. **Does this make the feature representations at fast timescales (2-4 words, 4-8 words) difficult to interpret?**

Because we separate embedding extraction from timescale interpretation, the use of a bidirectional language model does not make feature representations at fast timescales difficult to interpret. When extracting embeddings from the language model, the goal is to obtain embeddings that reflect the maximal amount of linguistic stimulus information. The filtering procedure for timescale interpretation is performed only after the word embeddings have been extracted; therefore the bidirectionality of the model does not prevent the interpretation of fast timescales.

Moreover, using the full-sentence context rather than a causal, rolling context window results in qualitatively similar estimates of timescale selectivity and better overall prediction performance; please see our response to Reviewer 2, Question 2 for more details.

2) Similarity, I'd like to see more justification for the choice of how the features were decomposed into separate frequency channels. **What is the advantage of the windowed-cosine approach over a simple Discrete Fourier Transform** (or is this a particular implementation of a DFT?)? The description of the approach says that it aims to find "components that vary with different periods" - **does this assume that the signals at each timescale will be periodic?**

The windowed-cosine approach is a particular implementation of the short-time Fourier transform (STFT). The STFT has the advantage of providing better temporal resolution to the extracted features (i.e., localizing each frequency band in the time-domain). The STFT does not assume that signals at each timescale are periodic.

3) The computation of the timescale selectivity value (section 4.5) makes sense, but seems like it biases preferred timescales toward the middle of the timescale range, since it involves a weighted average across all timescales. There is also a particular (implicit) choice being made about how to convert from correlations r to the selectivity profile - after setting negative values to 0, this is a softmax with temperature = 1. But other choices could be made here: for example, using a lower temperature would concentrate the selectivity profile on the timescale with maximum r value (which would make the profile more unstable, but decrease the bias toward the middle of the timescale range). **It would be helpful to see more justification of this step, and perhaps see how sensitive the results are to this choice of temperature.**

We used the mean to compute average timescale selectivity in order to incorporate information from each timescale in our estimate, and in order to produce more robust estimates of timescale selectivity – the averaging prevents small changes in prediction accuracy from producing large changes in estimated selectivity.

We have revised the manuscript to clarify why we use the weighted mean. The revised Methods states on Lines 524-526:

"We use the weighted average instead of simply taking the maximum selectivity across timescales, in order to prevent small changes in prediction accuracy from producing large changes in estimated timescale selectivity."

Note that estimates of timescale selectivity derived from taking the mean or the max across timescales are highly correlated. For each participant and modality, the correlation across voxels between estimates derived from the two temperature parameters is greater than 0.85. Nevertheless, when using the max across timescales, relatively small changes in prediction accuracy can produce larger changes in estimated timescales. To prevent these disproportionate changes, we therefore use the weighted average.

4) The way that Figure 3 is introduced is difficult to understand: "a voxel with the same timescale selectivity for reading and listening could nevertheless have different selectivities to *individual* timescales between the two modalities" (lines 104-106). After reading through the Methods, I believe this is saying that Figs 1+2 are describing the timescale selectivity of each voxel with a single number (a weighted average of all timescales, weighted by the selectivity to each timescale) while Figure 3 is separately correlating the selectivity to each individual timescale. Is this the correct interpretation?

Yes, this interpretation is correct. To clarify this point, we have revised the manuscript to clarify the distinction between voxels for which the selectivity to each timescale is peaked at one timescale, and voxels for which selectivity is uniform across several timescales.

The original manuscript stated on Lines 104-106:

"However, one potential confound is that a voxel with the same timescale selectivity for reading and listening could nevertheless have different selectivities to individual timescales between the two modalities."

The revised Results states on Lines 135-138:

"However, average timescale selectivity could equate two types of voxels: voxels with a very peaked selectivity for a single frequency band, and voxels with uniform selectivity to many frequency bands. To investigate this possibility we used the timescale selectivity profile, which reflects selectivity to each timescale separately."

5) The fact that the same stories were used for reading and listening means that it should be possible to simply compute the timecourse correlation between the responses to the two modalities for each story, for each voxel. This would be helpful to see, since it would provide a model-free measure of response similarity across modalities, potentially identifying voxels in which the current model fails to find similar timescales across voxels but a different model could.

Directly correlating timescales of BOLD responses is indeed an interesting analysis, and Regev et al. (2013) performed such an analysis. Regev et al. (2013) directly correlated timecourses of BOLD responses between reading and listening. They found that BOLD responses are highly correlated between modalities across some brain areas (e.g., pSTG, precuneus, angular gyrus, areas of prefrontal cortex), and differed in other brain areas (e.g., early sensory cortices, and parts of parietal and frontal cortices).

However, directly correlating timecourses of BOLD responses does not specifically model representations of linguistic information. Therefore, it is difficult to interpret results of timecourse-correlation in terms of language representation. Indeed, Regev et al. (2013) suggest that some differences between modalities may indicate differences between cognitive control processes, rather than differences language representations. Because we sought to specifically investigate integration of linguistic information, we specifically modeled brain representations of linguistic information rather than to directly correlate BOLD timecourses.

To clarify this distinction we have revised our manuscript. The revised Discussion now states on Lines 190-199:

"Other studies have observed similarities and differences in brain responses between spoken and written language comprehension. For instance, Regev et al. (2013) correlated brain responses between reading and listening. This work found similar representations in areas such as superior temporal gyrus, inferior frontal gyrus, and precuneus;

and different representations in early sensory areas as well as in parts of parietal and frontal cortices. However, Regev et al. (2013) did not specifically model linguistic features. They hypothesize that differing representations in higher-order areas may indicate distinct control processes rather than differences in representations of language itself. In contrast, our study specifically model linguistic features and found that these linguistic features are represented similarly between reading and listening. Our results suggest that some of the differences observed in (Regev et al., 2013) could indeed be due to extra-linguistic processes.”

6) **How is “Explainable Variance” computed in Supplementary Figure 6?**

“Explainable variance” is the fraction of variance in BOLD responses that is consistent across repetitions of the same stimulus. Mathematically, the explainable variance can be computed for a voxel with BOLD activity $Y \in \mathbb{R}^{N \times T}$ recorded over N runs in response to a stimulus with T TRs (note that Y must be zscored across time):

$$\begin{aligned} \text{residual} &= Y - \text{Mean}(Y, \text{axis} = 0) \\ \text{residual_variance} &= \text{Mean}(\text{Variance}(\text{residual}, \text{axis} = 1), \text{axis} = 0) \\ \text{ev} &= 1 - \text{residual_variance} \\ \text{bias_corrected_ev} &= \text{ev} - \frac{1 - \text{ev}}{N - 1} \end{aligned}$$

We have revised the manuscript to include this description alongside Supplementary Figure S2, which shows the explainable variance for each participant and modality.

Minor comments:

7) “Timescale” here is defined in terms of a number of words, rather than time in seconds. These will obviously be highly correlated, but **is it possible to instead apply the filters in the time (seconds) domain? Would there be any differences compared to the word-timescale hierarchy?**

We expect that estimates of voxel timescale selectivity would be highly correlated, whether defined in terms of number of words or in terms of number of seconds. Nevertheless, an interesting direction of future work would be to compare timescale selectivity defined in different units. It could be especially interesting to use speech language models (e.g., Whisper (Radford et al., 2023) or wav2vec 2.0 (Baevski et al., 2020)) to compute timescale selectivity in terms of seconds. To facilitate such comparisons, we release our feature extraction code. We have added a “Code Availability” section on Lines 567-570 with a link to our code.

8) **In section 4.4.4, I couldn’t understand exactly which feature spaces and delays are going into the embeddings.** There is a 9984-dimensional representation of each word, which is represented in 8 timescale filters, each delayed by 4 FIR delays? Are the sensory-level feature spaces (from 4.4.2) also being included in the regression? Are both sensory-level spaces used for both modalities, or does each modality get its own specific sensory-level features? Each feature space is given a separate regularization parameter - what does that mean here (e.g. are the different FIR delays different feature spaces)? The total number of feature spaces appears to be 10 based on section 4.4.5, so maybe the feature spaces are the 8 timescales and the 2 sensory-level features?

Yes, we use 8 filtered 9984-dimensional representation for each word. These representations are reflected in 8 filtered (9984 x num_words) matrices, each delayed by 4 FIR delays. The total number of 10 features does refer to the 8 timescale-specific feature spaces and the two sensory-level feature spaces. Both sensory-level spaces are used for both modalities. For both modalities, a separate regularization parameter is used for each voxel, feature space, and delay.

We have clarified these details in the text.

The original Methods stated on lines 364-365:

“The models for all feature spaces were jointly estimated in order to account for potential complementarity between feature spaces.”

The revised Methods now states on Lines 444-449:

"In order to account for potential complementarity between feature spaces, the models for all ten feature spaces were jointly estimated (Nunez-Elizalde et al., 2019; Dupré la Tour et al., 2022). These ten feature spaces comprise the eight timescale-specific feature spaces, and the two sensory-level feature spaces (the two sensory-level feature spaces reflect spectrotemporal features of the auditory stimulus and motion energy features of the visual stimulus.)"

The original Methods stated on Lines 373-375:

"Then banded ridge regression was used to estimate a mapping B (dimension $v \times (\sum_{i=1}^f p)$) from $F'(X)$ to the matrix of voxel responses Y (dimension $v \times n$). B is estimated according to $\hat{B} = \arg \min_B \|Y - BF'(X)\|_2^2 + \lambda \|CB\|_2^2$."

The revised Methods now states on Lines 456-459:

"Then banded ridge regression was used to estimate a mapping B (dimension $v \times (\sum_{i=1}^f p)$) from $F'(X)$ to the matrix of voxel responses Y (dimension $v \times n$). B is estimated according to $\hat{B} = \arg \min_B \|Y - BF'(X)\|_2^2 + \lambda \|CB\|_2^2$. A separate regularization parameter was fit for each voxel, feature space, and FIR delay."

9) **Due to the 120s high-pass filter applied during preprocessing, there should be some rough upper limit to the longest-timescale feature space** - is there an approximate upper limit of the 256+ word filter?

Indeed, the low-pass filtering used to remove low-frequency voxel response drift may also remove very long-timescale representations. The low-pass filter consisted of a Savitsky-Golay filter with a window size of 120s, and the stimuli contain an average of 3 words per second. Therefore there is an approximate upper limit of 360 words in the preprocessed data. This upper limit could explain why the 256+ word feature space produces poor prediction performance relative to other timescale-specific feature spaces.

To clarify this limitation, we have revised the Discussion to include the following description on Lines 243-245:

"Furthermore, during preprocessing the BOLD data were low-pass filtered in order to control for low-frequency voxel response drift. This preprocessing filter may have removed information about brain representations of very long timescales (i.e., timescales above 360 words)."

10) Permutation testing was performed by block shuffling the target timecourse in blocks of 10 TRs. This is a reasonable approach (and has precedent in other work in this field) but the block length seems relatively short compared to the relevant timescales in this study (which go up to the paragraph+ scale). **Would an alternative approach like circularly-shifting the target timecourse be more appropriate?**

We shuffle in blocks of 10 TRs to account for the hemodynamic response function (HRF) of voxel BOLD responses. Circularly-shifting the target timecourse would result in fewer independent null distribution points – e.g., shuffling the target timecourse by 10 TRs would be highly similar to shuffling the target timecourse by 11 TRs. In order to generate a sufficient number of null distribution points, we therefore performed block shuffling on the target timecourse.

11) **The authors do not provide a "Code Availability" statement/section.** The technical approach used here is central to the paper, and it would be helpful (both for reproducibility as well as for other researchers hoping to apply this method to their own datasets) if the code could be publicly released, unless there is a compelling reason why this is not possible.

We have made our feature extraction code available at https://github.com/denizenslab/timescales_filtering, and we have added a "Code Availability" section on Lines 567-570 with a link to our code.

12) In Figure 1, at first I mistakenly thought that panels a and b were group data - **the authors could make it more clear that these are results from S1.**

We apologize for the unclear presentation. We have revised the manuscript to more clearly indicate that Panels A and B of Figure 1 are from S1.

13) There is a type-setting error on line 387: the 10^{10} exponent is not properly rendered.

We thank the reviewer for pointing out this typo. We have revised the manuscript to fix the rendering of this exponent.

References

- Baevski, A., Zhou, Y., Mohamed, A., and Auli, M. (2020). wav2vec 2.0: A framework for self-supervised learning of speech representations. *Advances in neural information processing systems*, 33:12449–12460.
- Baldassano, C., Chen, J., Zadbood, A., Pillow, J. W., Hasson, U., and Norman, K. A. (2017). Discovering event structure in continuous narrative perception and memory. *Neuron*, 95(3):709–721.
- Blank, I. and Fedorenko, E. (2020). No evidence for differences among language regions in their temporal receptive windows. *NeuroImage*, 219.
- Booth, J. R., Burman, D. D., Meyer, J. R., Gitelman, D. R., Parrish, T. B., and Mesulam, M. M. (2002). Modality independence of word comprehension. *Human brain mapping*, 16(4):251–261.
- Buchweitz, A., Mason, R. A., Tomitch, L., and Just, M. A. (2009). Brain activation for reading and listening comprehension: An fmri study of modality effects and individual differences in language comprehension. *Psychology & neuroscience*, 2:111–123.
- Caucheteux, C. and King, J.-R. (2020). Language processing in brains and deep neural networks: computational convergence and its limits. *BioRxiv*.
- de Heer, W. A., Huth, A. G., Griffiths, T. L., Gallant, J. L., and Theunissen, F. E. (2017). The hierarchical cortical organization of human speech processing. *Journal of Neuroscience*, 37(27):6539–6557.
- Deniz, F., Nunez-Elizalde, A., Huth, A. G., and Gallant, J. (2019). The representation of semantic information across human cerebral cortex during listening versus reading is invariant to stimulus modality. *The Journal of Neuroscience*, 39:7722 – 7736.
- Deniz, F., Tseng, C., Wehbe, L., la Tour, T. D., and Gallant, J. L. (2023). Semantic representations during language comprehension are affected by context. *Journal of Neuroscience*, 43(17):3144–3158.
- Dupré la Tour, T., Eickenberg, M., Nunez-Elizalde, A. O., and Gallant, J. L. (2022). Feature-space selection with banded ridge regression. *NeuroImage*, 264:119728.
- Heilbron, M., Richter, D., Ekman, M., Hagoort, P., and De Lange, F. P. (2020). Word contexts enhance the neural representation of individual letters in early visual cortex. *Nature communications*, 11(1):1–11.
- Jain, S. and Huth, A. (2018). Incorporating context into language encoding models for fMRI. In *Advances in Neural Information Processing Systems*, volume 31, pages 6628–6637.
- Jain, S., Vo, V. A., Mahto, S., LeBel, A., Turek, J. S., and Huth, A. G. (2020). Interpretable multi-timescale models for predicting fMRI responses to continuous natural speech. In *Advances in Neural Information Processing Systems*.
- Lerner, Y., Honey, C. J., Silbert, L. J., and Hasson, U. (2011). Topographic mapping of a hierarchy of temporal receptive windows using a narrated story. *Journal of Neuroscience*, 31(8):2906–2915.
- Li, M., Lv, T., Chen, J., Cui, L., Lu, Y., Florencio, D., Zhang, C., Li, Z., and Wei, F. (2023). Trocr: Transformer-based optical character recognition with pre-trained models. In *Proceedings of the AAAI Conference on Artificial Intelligence*, volume 37, pages 13094–13102.

- Liuzzi, A. G., Bruffaerts, R., Peeters, R., Adamczuk, K., Keuleers, E., De Deyne, S., Storms, G., Dupont, P., and Vandenberghe, R. (2017). Cross-modal representation of spoken and written word meaning in left pars triangularis. *Neuroimage*, 150:292–307.
- Millet, J., Caucheteux, C., Boubenec, Y., Gramfort, A., Dunbar, E., Pallier, C., King, J.-R., et al. (2022). Toward a realistic model of speech processing in the brain with self-supervised learning. *Advances in Neural Information Processing Systems*, 35:33428–33443.
- Nakai, T., Yamaguchi, H. Q., and Nishimoto, S. (2021). Convergence of modality invariance and attention selectivity in the cortical semantic circuit. *Cerebral Cortex*, 31(10):4825–4839.
- Nunez-Elizalde, A. O., Huth, A. G., and Gallant, J. L. (2019). Voxelwise encoding models with non-spherical multivariate normal priors. *Neuroimage*, 197:482–492.
- Oota, S. R., Arora, J., Agarwal, V., Marreddy, M., Gupta, M., and Surampudi, B. (2022). Neural language taskonomy: Which NLP tasks are the most predictive of fMRI brain activity? In *Proceedings of the 2022 Conference of the North American Chapter of the Association for Computational Linguistics: Human Language Technologies*, pages 3220–3237, Seattle, United States. Association for Computational Linguistics.
- Radford, A., Kim, J. W., Xu, T., Brockman, G., McLeavey, C., and Sutskever, I. (2023). Robust speech recognition via large-scale weak supervision. In *International Conference on Machine Learning*, pages 28492–28518. PMLR.
- Regev, M., Honey, C. J., Simony, E., and Hasson, U. (2013). Selective and invariant neural responses to spoken and written narratives. *Journal of Neuroscience*, 33(40):15978–15988.
- Schrimpf, M., Blank, I. A., Tuckute, G., Kauf, C., Hosseini, E. A., Kanwisher, N., Tenenbaum, J. B., and Fedorenko, E. (2021). The neural architecture of language: Integrative modeling converges on predictive processing. *Proceedings of the National Academy of Sciences*, 118(45):e2105646118.
- Simony, E., Honey, C. J., Chen, J., Lositsky, O., Yeshurun, Y., Wiesel, A., and Hasson, U. (2016). Dynamic reconfiguration of the default mode network during narrative comprehension. *Nature communications*, 7(1):12141.
- Toneva, M. and Wehbe, L. (2019). Interpreting and improving natural-language processing (in machines) with natural language-processing (in the brain). In *Advances in Neural Information Processing Systems*.
- Yeo, B. T., Krienen, F. M., Sepulcre, J., Sabuncu, M. R., Lashkari, D., Hollinshead, M., Roffman, J. L., Smoller, J. W., Zöllei, L., Polimeni, J. R., et al. (2011). The organization of the human cerebral cortex estimated by intrinsic functional connectivity. *Journal of neurophysiology*.

Reviewers' comments:

Reviewer #1 (Remarks to the Author):

We would like to express our sincere appreciation for the considerable effort the authors have put into revising the manuscript based on our initial feedback. The amendments made have significantly strengthened the paper, and we commend the authors for their dedication to improving the quality of their work. However, while most of our concerns have been effectively addressed, two of the analyses still need to be clarified with additional statistical tests and figures.

Stimulus embedding:

In figure R1.A., the authors show the number of significantly predicted voxels (i.e., language-selective voxels) for each layer of BERT, with the claim that better predictions of brain responses come from the stimulus embedding method that includes all layers versus any individual layer. However, similarities in the amount of "language-selective voxels" (i.e., voxels predicted by any of the eight timescale-specific feature spaces) for each layer is not congruent with the claim that these maps are similar to each other, layer by layer, with respect to the distribution of timescale-selective voxels. Figure R1.B. qualitatively addresses this by showing the timescale selectivity estimates for each layer on the flattened cortical surfaces of participants S1 and S2 (R1.B). And while we agree that that the maps are suggestive of timescale selectivity being robust to the choice of stimulus embedding for both reading and listening, the extent of convergence needs to be investigated more systematically and include all participants. We propose the authors calculate the voxelwise spatial correlation of timescale selectivity across stimulus embedding methods (i.e., for each layer and the concatenated embedding) for each participant and for each stimulus modality. Correlation matrices for both reading and listening can be constructed, such that each row and column corresponds to a layer, with values being the (dis)similarity averaged across participants. Only then can the similarity of the layer vs. full-model predictions for the different timescales be assessed, which is important for understanding how the layers relate to the timescale selectivity results.

It will also be important to the reader to have a clear discussion of how the results in this manuscript relate to previous studies that find that timescale selectivity varies by layer in BERT. Specifically, the authors should cite these references in their discussion: Jawahar et al., 2019; Tenney et al., 2019, Rogers et al., 2020.

Jawahar, G., Sagot, B., & Seddah, D. (2019, July). What does BERT learn about the structure of language? In ACL 2019-57th Annual Meeting of the Association for Computational Linguistics.

Niu, J., Lu, W., & Penn, G. (2022, October). Does BERT Rediscover a Classical NLP Pipeline?. In Proceedings of the 29th International Conference on Computational Linguistics(pp. 3143-3153).

Rogers, A., Kovaleva, O., & Rumshisky, A. (2021). A primer in BERTology: What we know about how BERT works. Transactions of the Association for Computational Linguistics, 8, 842-866.

Sequence input context length:

Similar to the above, the authors need to calculate the voxelwise spatial correlation between maps derived from different context lengths for each modality and for each participant. Again, the side-by-side maps of S1 and S2 suggest that the topographic organization of timescale selectivity is robust to the choice of input context length, but this claim needs to be statistically supported and include all participants.

Reviewer #2 (Remarks to the Author):

I thank the authors for the response. It is clear they took my concerns seriously and thoroughly addressed my feedback. I am happy with the new analyses that the authors have added related to the robustness to different ways of extracting the word embeddings, the entropy of the timescale selectivity, and the voxel-wise correlations of the timescale selectivity profiles between reading & listening.

I have just one thought regarding the additional analysis with extracting the word embeddings using a fixed-length context window. This is strictly a curiosity and not something that I expect to be investigated in this manuscript. The authors observe that the brain activity is better predicted using the original way of extracting the word representations using the context of the whole sentence (including words before and after the current word). There are several differences between the two approaches of word embedding extraction: 1) the amount of context the model is using to create the representation, 2) whether the model has access to previous and future context, 3) whether the context is within sentence or not. It's not clear which of these reasons is the biggest contributor to the sentence-context representations performing better at brain prediction -- my guess is that the reason is 2). One could imagine that since the participants listen to the story twice, they may actually be better at predicting upcoming information during the second listen. I wonder whether the difference the authors observe between the brain prediction performance for the sentence-context and fixed-context representations is larger during the second listen than the first, which may provide some evidence for increased prediction. Though of course this evidence would be strengthened by showing that the other two possible reasons for the difference are not the main contributor.

Reviewer #3 (Remarks to the Author):

The authors have thoroughly addressed my concerns in the revision. I have only one remaining clarification on my comments I wanted to add:

1c) My concern about interpreting fast-timescale representations is that word representations in a bidirectional language model can be influenced by words much later in the sentence. For example, in the sentences "The bat that I keep in my attic is very special to me, since for my whole life I have always liked

baseball" and "The bat that I keep in my attic is very special to me, since for my whole life I have always liked *animals*", the representation of the word "bat" will be different. Even when filtering at a fast timescale that only considers the first two words of this sentence, this filtered embedding would contain information from words at the end of the sentence. The fact that the causal context window yields similar results suggests that this is not a major issue, but could be worth clarifying in the paper.

Response to reviewer's comments: The Cortical Representation of Language Timescales is Shared between Reading and Listening

We thank all reviewers for their positive comments, and we are glad that our revisions have addressed most of their concerns. We revised our manuscript according to the reviewers' remaining comments and suggestions, and include point-by-point responses to each reviewer's questions.

1 Reviewer #1 (Remarks to the Author):

We would like to express our sincere appreciation for the considerable effort the authors have put into revising the manuscript based on our initial feedback. The amendments made have significantly strengthened the paper, and we commend the authors for their dedication to improving the quality of their work. However, while most of our concerns have been effectively addressed, two of the analyses still need to be clarified with additional statistical tests and figures.

Stimulus embedding:

In figure R1.A., the authors show the number of significantly predicted voxels (i.e., language-selective voxels) for each layer of BERT, with the claim that better predictions of brain responses come from the stimulus embedding method that includes all layers versus any individual layer. However, similarities in the amount of "language-selective voxels" (i.e., voxels predicted by any of the eight timescale-specific feature spaces) for each layer is not congruent with the claim that these maps are similar to each other, layer by layer, with respect to the distribution of timescale-selective voxels. Figure R1.B. qualitatively addresses this by showing the timescale selectivity estimates for each layer on the flattened cortical surfaces of participants S1 and S2 (R1.B). And while we agree that that the maps are suggestive of timescale selectivity being robust to the choice of stimulus embedding for both reading and listening, the extent of convergence needs to be investigated more systematically and include all participants. We propose the authors calculate the voxelwise spatial correlation of timescale selectivity across stimulus embedding methods (i.e., for each layer and the concatenated embedding) for each participant and for each stimulus modality. Correlation matrices for both reading and listening can be constructed, such that each row and column corresponds to a layer, with values being the (dis)similarity averaged across participants. Only then can the similarity of the layer vs. full-model predictions for the different timescales be assessed, which is important for understanding how the layers relate to the timescale selectivity results.

We thank the reviewer for their thoughtful comment to examine the embedding extraction methods more systematically. We agree that the similarity between the number of language-selective voxels is not congruent to the similarity in estimates of timescale selectivity. Following the reviewer's suggestion, we more systematically compared the effect of BERT layer on estimates of timescale selectivity. For each of the nine participants and two modalities, we fit 13 separate encoding models: one model for each layer of BERT. Each model was separately used to estimate timescale selectivity. We then computed for each stimulus modality the voxelwise spatial correlation of timescale selectivity between each pair of models, over the voxels that are significantly predicted by both models.

Figure R1 shows the group-averaged voxelwise spatial correlation between each pair of layers and for each stimulus modality. Estimates of timescale selectivity are highly correlated across layers ($r > 0.7$; $p < .05$ for each of nine participants, two modalities, and 98 pairwise layer comparisons, by one-sided permutation tests that were FDR corrected with a Benjamini-Hochberg cor-

Fig. R1: **Comparison of estimated timescale selectivity between different embedding layers.** Timescale selectivity was estimated separately with embeddings from each layer of BERT. Group-averaged voxelwise spatial correlation between timescale selectivity is shown for each pair of layers, for reading (A) and listening (B) separately. Estimates of timescale selectivity are highly correlated across layers. Estimates are more similar for layers that are closer together, suggesting a small effect of the stimulus layer on estimates of timescale selectivity. Overall, estimates of timescale selectivity are consistent across different layers.

rection for multiple comparisons). Estimates are more similar for layers that are closer together, suggesting that there is a small effect of stimulus layer on estimates of timescale selectivity. Overall, for both modalities, estimates of timescale selectivity are highly consistent across different embedding layers.

We have revised the manuscript to include Figure R1 as Supplementary Figure S11.

It will also be important to the reader to have a clear discussion of how the results in this manuscript relate to previous studies that find that timescale selectivity varies by layer in BERT. Specifically, the authors should cite these references in their discussion: Jawahar et al., 2019; Tenney et al., 2019, Rogers et al., 2020.

Jawahar, G., Sagot, B., & Seddah, D. (2019, July). What does BERT learn about the structure of language? In ACL 2019-57th Annual Meeting of the Association for Computational Linguistics.

Niu, J., Lu, W., & Penn, G. (2022, October). Does BERT Rediscover a Classical NLP Pipeline?. In Proceedings of the 29th International Conference on Computational Linguistics(pp. 3143-3153).

Rogers, A., Kovaleva, O., & Rumshisky, A. (2021). A primer in BERTology: What we know about how BERT works. Transactions of the Association for Computational Linguistics, 8, 842-866.

We have revised the manuscript to discuss how our results relate to previous studies of BERT. The revised Methods now states on page 12:

Prior work suggested that language structures with different timescales are preferentially represented in different layers of BERT (Tenney et al. (2019); Jawahar et al. (2019); Rogers et al. (2021); but see Niu et al. (2022) for an argument that language timescales are not cleanly separated across different layers of BERT). Earlier layers represent lower-level, shorter-timescale information (e.g., word identity and linear word order), whereas later layers represent higher-level, longer-timescale information (e.g., coreference, long-distance dependencies). To include stimulus information at all levels of the

language processing hierarchy, activations from all layers of BERT were included in the stimulus embedding.

Sequence input context length: Similar to the above, the authors need to calculate the voxelwise spatial correlation between maps derived from different context lengths for each modality and for each participant. Again, the side-by-side maps of S1 and S2 suggest that the topographic organization of timescale selectivity is robust to the choice of input context length, but this claim needs to be statistically supported and include all participants.

Following the reviewer's suggestion, we compared estimates of timescale selectivity across different input context lengths. For each of the nine participants and two modalities, we fit three separate encoding models: one model with sentence-length input contexts, one model with a rolling input context of 10 words, and one model with a rolling input context of 100 words. Each model was separately used to estimate timescale selectivity. We then computed for each stimulus modality the voxelwise spatial correlation of timescale selectivity between each pair of models over the voxels that are significantly predicted by both models.

Figure R2 shows the group-averaged voxelwise spatial correlation between each pair of stimulus embedding methods, for reading and listening. The sentence-length input context and rolling input context of 10 produce highly correlated estimates of timescale selectivity ($r = 0.65$). Estimates from longer rolling input contexts are less strongly correlated with estimates from the other two embedding methods ($r = 0.23, 0.26$). Correlations were statistically significant at $p < .05$ for each of nine participants, two modalities, and three pairwise input-length comparisons (by one-sided permutation tests that were FDR corrected with a Benjamini-Hochberg correction for multiple comparisons). We hypothesize that the lower correlation with the 100-word context model reflects the poorer prediction accuracy of the 100-word context length model (Shown in Fig. S7). These results suggest that using a shorter input context length produces accurate models of brain responses. Estimates of timescale selectivity are overall consistent between well-predicting stimulus embedding methods.

We have revised the manuscript to include figure R2 as Supplementary Figure S9.

Fig. R2: Comparison of estimated timescale selectivity between different input context lengths. Timescale selectivity was estimated separately with a sentence-length stimulus input context, a rolling 10-word input context, and a rolling 100-word input context. For each pair of input contexts, group-averaged voxelwise spatial correlation between estimated timescale selectivity is shown, for reading (A) and listening (B) separately. The sentence-length context and rolling 10-word context produce correlated estimates of timescale selectivity. The rolling 100-word context produces estimates of timescale selectivity that are less similar, but still positively correlated with estimates from the other two embedding methods. Furthermore, the similarity between sentence-length and rolling input contexts suggests that the inclusion of future context in sentence-length input contexts does not qualitatively change estimates of timescale selectivity.

2 Reviewer #2 (Remarks to the Author):

I thank the authors for the response. It is clear they took my concerns seriously and thoroughly addressed my feedback. I am happy with the new analyses that the authors have added related to the robustness to different ways of extracting the word embeddings, the entropy of the timescale selectivity, and the voxel-wise correlations of the timescale selectivity profiles between reading & listening.

I have just one thought regarding the additional analysis with extracting the word embeddings using a fixed-length context window. This is strictly a curiosity and not something that I expect to be investigated in this manuscript. The authors observe that the brain activity is better predicted using the original way of extracting the word representations using the context of the whole sentence (including words before and after the current word). There are several differences between the two approaches of word embedding extraction: 1) the amount of context the model is using to create the representation, 2) whether the model has access to previous and future context, 3) whether the context is within sentence or not. It's not clear which of these reasons is the biggest contributor to the sentence-context representations performing better at brain prediction – my guess is that the reason is 2). One could imagine that since the participants listen to the story twice, they may actually be better at predicting upcoming information during the second listen. I wonder whether the difference the authors observe between the brain prediction performance for the sentence-context and fixed-context representations is larger during the second listen than the first, which may provide some evidence for increased prediction. Though of course this evidence would be strengthened by showing that the other two possible reasons for the difference are not the main contributor.

We thank the reviewer for their comment. To address the reviewer's question, we explored whether there is evidence for increased prediction for the second vs the first presentation of the test story. We compared the prediction accuracy of sentence-length contexts vs a rolling input context of 10 words, separately for the first and second presentation of the test story (the modality of stimulus comprehension was the same for the first two presentations of the test run).

The prediction accuracy for the two embedding methods is shown for each participant in Figure R3, separately for the first and second presentation of the test story. For both presentations of the test story, sentence contexts produce more accurate predictions than rolling contexts. Interestingly, the improvement of sentence contexts is *greater* for the first presentation of the test story. This improvement suggests that the higher accuracy of sentence contexts may be driven more by the use of within-sentence inputs, rather than by access to future context.

Fig. R3: **Comparison of sentence vs rolling input context, first vs second test presentation.** For each participant and stimulus embedding method, prediction accuracy was evaluated separately for the first and second presentations of the test story. Prediction accuracy is shown for a sentence-length context (x-axis) vs a rolling 10-word context (y-axis). Predictions are shown for each participant, for the first and second presentation of the test story separately. Sentence contexts produce more accurate predictions than rolling contexts for both presentations of the test story. Improvement of sentence contexts is greater for the first than the second presentation of the test story. Thus, higher accuracy of sentence contexts may be driven more by the use of within-sentence inputs than by access to future context.

3 Reviewer #3 (Remarks to the Author):

The authors have thoroughly addressed my concerns in the revision. I have only one remaining clarifications on my comments I wanted to add:

1c) My concern about interpreting fast-timescale representations is that word representations in a bidirectional language model can be influenced by words much later in the sentence. For example, in the sentences "The bat that I keep in my attic is very special to me, since for my whole life I have always liked *baseball*" and "The bat that I keep in my attic is very special to me, since for my whole life I have always liked *animals*", the representation of the word "bat" will be different. Even when filtering at a fast timescale that only considers the first two words of this sentence, this

filtered embedding would contain information from words at the end of the sentence. The fact that the causal context window yields similar results suggests that this is not a major issue, but could be worth clarifying in the paper.

We agree that similarities between estimates from the rolling (causal) context and the sentence-length input context suggests that this is not a major issue. We think that this point is worth clarifying, and we have added a clarification in the paper. The new Supplementary Figure S9 was added in our response to Reviewer 1. This new analysis that more rigorously shows that the timescale selectivity is similar between the 10-word causal context and the sentence-length input context (Figure R2, included as Supplementary Figure S9 in the revised manuscript.) In the caption of this figure we clarify the point about future contexts. The caption of the new Supplementary Figure S9 states:

Furthermore, the similarity between sentence-length and rolling input contexts suggests that the inclusion of future context in sentence-length input contexts does not qualitatively change estimates of timescale selectivity.

References

- Jawahar, G., Sagot, B., and Seddah, D. (2019). What does bert learn about the structure of language? In *ACL 2019-57th Annual Meeting of the Association for Computational Linguistics*.
- Niu, J., Lu, W., and Penn, G. (2022). Does bert rediscover a classical nlp pipeline? In *Proceedings of the 29th International Conference on Computational Linguistics*, pages 3143–3153.
- Rogers, A., Kovaleva, O., and Rumshisky, A. (2021). A primer in bertology: What we know about how bert works. *Transactions of the Association for Computational Linguistics*, 8:842–866.
- Tenney, I., Das, D., and Pavlick, E. (2019). Bert rediscovers the classical nlp pipeline. In *Annual Meeting of the Association for Computational Linguistics*.

REVIEWERS' COMMENTS:

Reviewer #1 (Remarks to the Author):

We appreciate the authors' efforts to respond to our comments and believe that the manuscript is now ready for publication. Our only minor suggestions are:

1) For Supplementary Figures 9 and 11, it would help the reader if the color scale used was more standard, ie, that brightness corresponded to proximity to the ends of the scale (-1, 1) rather than 0. That way greater correlation is brighter. This is the case for any other figures that use this color scale.

2) On page 2 line 90, the authors use the word "qualitatively" to describe the similarity of short vs. sentence-length timescales. The authors should use "quantitatively" instead to give themselves the credit for doing the analyses in the indicated Supplementary figures.